# Reducing Blackwell and Average Optimality to Discounted MDPs via the Blackwell Discount Factor

**Julien Grand-Clément**
ISOM Department
HEC Paris
grand-clement@hec.fr

**Marek Petrik**
Department of Computer Science
University of New Hampshire
mpetrik@cs.unh.edu

## Abstract

We introduce the *Blackwell discount factor* for Markov Decision Processes (MDPs). Classical objectives for MDPs include discounted, average, and Blackwell optimality. Many existing approaches to computing average-optimal policies solve for discount-optimal policies with a discount factor close to 1, but they only work under strong or hard-to-verify assumptions such as unichain or ergodicity. We highlight the shortcomings of the classical definition of Blackwell optimality, which does not lead to simple algorithms for computing Blackwell-optimal policies and overlooks the pathological behaviors of optimal value functions with respect to the discount factors. To resolve this issue, we show that when the discount factor is larger than the *Blackwell discount factor* $\gamma_{\mathsf{bw}}$, all discount-optimal policies become Blackwell- and average-optimal, and we derive a general upper bound on $\gamma_{\mathsf{bw}}$. Our upper bound on $\gamma_{\mathsf{bw}}$, parametrized by the *bit-size* of the rewards and transition probabilities of the MDP instance, provides the first reduction from average and Blackwell optimality to discounted optimality, *without any assumptions*, along with new polynomial-time algorithms. Our work brings new ideas from polynomials and algebraic numbers to the analysis of MDPs. Our results also apply to robust MDPs, enabling the first algorithms to compute robust Blackwell-optimal policies.

## 1 Introduction

Markov Decision Processes (MDPs) provide a widely-used framework for modeling sequential decision-making problems (Puterman, 2014). In a (finite) MDP, the decision maker repeatedly interacts with an environment characterized by a finite set of states and a finite set of available actions. The decision maker follows a *policy* that prescribes an action at a state at every period. An instantaneous reward is obtained at every period, depending on the current state-action pair, and the system transitions to the next state at the next period. MDPs provide the underlying model for the applications of reinforcement learning (RL), ranging from healthcare (Gottesman et al., 2019) to game solving (Mnih et al., 2013) and finance (Deng et al., 2016).

There are several optimality criteria that measure a decision maker's performance in an MDP. In *discounted optimality*, the decision maker optimizes the discounted return, defined as the sum of the instantaneous rewards over the infinite horizon, where future rewards are discounted with a *discount factor* $\gamma \in [0, 1)$. In *average optimality*, the decision maker optimizes the average return, defined as the average of the instantaneous rewards obtained over the infinite horizon. The average return ignores any return gathered in finite time, i.e., it does not reflect the transient performance of a policy and it only focuses on the steady-state behavior. The most selective optimality criterion in MDPs is *Blackwell optimality* (Puterman, 2014). A policy is Blackwell-optimal if it optimizes the discounted return simultaneously for all discount factors sufficiently close to 1. Since a discount factor close

37th Conference on Neural Information Processing Systems (NeurIPS 2023).

to 1 can be interpreted as a preference for rewards obtained in later periods, Blackwell-optimal policies are also average-optimal. However, average-optimal policies need not be Blackwell-optimal. Blackwell optimality can be a useful criterion in environments with no natural, or known, discount factor. Also, any algorithm that computes a Blackwell-optimal policy also immediately computes an average-optimal policy. This is one of the reasons why better understanding the Blackwell optimality criterion is mentioned as *"one of the pressing questions in RL"* in the list of open research problems from a recent survey on RL for average reward optimality (Dewanto et al., 2020).

Average-optimal policies can be computed via linear programming (section 9.3, (Puterman, 2014)). However, virtually all of the recent algorithms for computing average-optimal policies require strong assumptions on the underlying Markov chains associated with the policies in the MDP instance, such as ergodicity (Wang, 2017), the unichain and aperiodicity properties (Schneckenreither, 2020), weakly communicating MDPs (Wang et al., 2022), or assumptions on the mixing time associated with any deterministic policies (Jin and Sidford, 2020, 2021). These assumptions are motivated by technical considerations (e.g., ensuring that the average reward is uniform across all states) and can be restrictive in practice (Puterman, 2014) and NP-hard to verify, such as unichain (Tsitsiklis, 2007). Existing methods for computing Blackwell-optimal policies rely on linear programming over the field of power series including negative coefficients (Hordijk et al., 1985), or on an algorithm based on a nested sequence of optimality equations (O'Sullivan and Veinott Jr, 2017) which requires to solve multiple linear programs sequentially. These algorithms are complex, difficult to implement, and have no complexity guarantees or known implementations.

In summary, existing algorithms for average optimality require restrictive assumptions, and algorithms for Blackwell-optimality are very complex. This is in stark contrast with the vast literature on solving discounted MDPs, where general and well-understood methods exist, including value iteration, policy iteration, and linear programming (chapter 6, (Puterman, 2014)). This is the starting point of this paper, which aims to develop new algorithms for computing average-optimal and Blackwell-optimal policies through a reduction to discounted MDPs. We make the following **three main contributions.**

*1) A new definition of Blackwell optimality via the Blackwell discount factor* $\gamma_{\mathsf{bw}} \in [0, 1)$. Our first main contribution is to highlight that the standard definition of Blackwell optimality cannot be used to compute Blackwell-optimal policies with simple algorithms. Standard definitions have focused on *necessary* condition for Blackwell optimal policies to be discount optimal. However, we show that this condition needs to be revised when one seeks to compute a Blackwell-optimal policy. We do so by highlighting the potential pathological behaviors of the value functions: a Blackwell-optimal policy may be optimal on an arbitrary number of arbitrary disjoint intervals, and other non-Blackwell optimal policies may also be discount-optimal for some discount factors very close to 1. Demonstrating this issue is important because previous literature has repeatedly overlooked it. To address this issue, we introduce and show the existence of a discount factor $\gamma_{\mathsf{bw}}$ such that discount optimality for $\gamma > \gamma_{\mathsf{bw}}$ is *sufficient* for Blackwell optimality. Knowing the discount factor $\gamma_{\mathsf{bw}}$ is vital because it enables one to compute Blackwell- and average-optimal policies simply by solving a discounted MDP with $\gamma \in (\gamma_{\mathsf{bw}}, 1)$, for which there exist well-studied, simple, and efficient algorithms.

*2) Upper-bound the Blackwell discount factor.* As our second main contribution, we provide a strict upper bound on $\gamma_{\mathsf{bw}}$ given an MDP instance. We show that an upper bound must depend on $\boldsymbol{r}$ and $\boldsymbol{P}$, and we compute a bound that is parametrized by the number of states and the number of bits required to represent the MDP instance. Solving a discounted MDP with a discount factor larger or equal than our strict upper bound returns a Blackwell-optimal policy. Crucially, *our strict upper bound requires no assumptions on the underlying structure of the MDP*, which is a significant improvement on existing literature. Interestingly, the construction of our upper bound relies on novel techniques for analyzing MDPs. We interpret $\gamma_{\mathsf{bw}} \in [0, 1)$ as the root of a polynomial equation $p(\gamma) = 0$ in $\gamma$, show $p(1) = 0$, and use a lower bound $\mathsf{sep}(p)$ on the distance between any two roots of a polynomial $p$, known as the *separation of algebraic numbers*. This shows that $\gamma_{\mathsf{bw}} < 1 - \mathsf{sep}(p)$, where $\mathsf{sep}(p)$ depends on the MDP instance. Since Blackwell optimality implies average optimality, we also obtain the first reduction from average optimality to discounted optimality, *without any assumption* on the MDP structure. Our upper bound on $\gamma_{\mathsf{bw}}$ is itself of polynomial size in the bit-size of the MDP data. Combining this bound with interior-point methods for solving discounted MDPs, we obtain new weakly-polynomial time algorithms for computing Blackwell-optimal and average-optimal policies.

*3) Blackwell discount factor for robust MDPs.* We consider the case of robust reinforcement learning where the transition probabilities are unknown and, instead, belong to an uncertainty set. As our

third main contribution, we show that the robust Blackwell discount factor $\gamma_{\text{bw,r}}$ exists for popular models of uncertainty, such as sa-rectangular robust MDPs with polyhedral uncertainty (Goyal and Grand-Clément, 2023b, Iyengar, 2005). For this setting, we generalize our upper bound on $\gamma_{\text{bw}}$ for MDPs to an upper bound on $\gamma_{\text{bw,r}}$ for robust MDPs. Since robust MDPs with discounted optimality can be solved via value iteration and policy iteration, we provide the very first algorithms to compute Blackwell-optimal policies for robust MDPs.

We conclude this section with a discussion on **related works**. Several papers study the reduction of average optimality policy to discounted optimality under strong assumptions. Early attempts include (Ross, 1968), assuming that all transition probabilities are lower bounded by $\epsilon > 0$. Recent extensions assume bounded times of first returns (Akian and Gaubert, 2013, Huang, 2016), or weakly-communicating MDPs (Wang et al., 2022). Note that checking that an MDP instance is weakly-communicating can be done in polynomial-time (Kallenberg, 2002), in contrast to the unichain assumption (Tsitsiklis, 2007). The case of deterministic MDPs is treated in (Friedmann, 2011, Perotto and Vercouter, 2018, Zwick and Paterson, 1996). Other reductions require assumptions on the mixing times of the Markov chains induced by deterministic policies (Jin and Sidford, 2021). (Boone and Gaujal, 2022) propose a sampling algorithm to learn a Blackwell-optimal policy, in a special case in which it reduces to bias optimality. Under the condition that the robust MDP is unichain and that there is a unique average optimal policy, (Wang et al., 2023) show the existence of Blackwell-optimal policies for sa-rectangular robust MDPs, which is connected to the existence results in (Tewari and Bartlett, 2007) and (Goyal and Grand-Clément, 2023b) for polyhedral uncertainty. In contrast to the existing literature, one of the core strengths of our results is that we do not need any structural assumption on the Markov chains of the underlying MDP to obtain our reduction from Blackwell optimality and average optimality to discounted optimality.

## 2 Preliminaries on MDPs

An MDP instance is characterized by a tuple $\mathcal{M} = (\mathcal{S}, \mathcal{A}, \boldsymbol{r}, \boldsymbol{P})$, where $\mathcal{S}$ is a finite set of states and $\mathcal{A}$ is a finite set of actions. The instantaneous rewards are denoted by $\boldsymbol{r} \in \mathbb{R}^{\mathcal{S} \times \mathcal{A}}$ and the transition probabilities are denoted by $\boldsymbol{P} \in (\Delta(\mathcal{S}))^{\mathcal{S} \times \mathcal{A}}$, where $\Delta(\mathcal{S})$ is the simplex over $\mathcal{S}$. At any time period $t$, the decision maker is in a state $s_t \in \mathcal{S}$, chooses an action $a_t \in \mathcal{A}$, obtains an instantaneous reward $r_{s_t a_t} \in \mathbb{R}$, and transitions to state $s_{t+1}$ with probability $P_{s_t a_t s_{t+1}} \in [0, 1]$. A *deterministic stationary* policy $\pi \colon \mathcal{S} \to \mathcal{A}$ assigns an action to each state. Importantly, there exists an optimal deterministic stationary policy for all the criteria considered in this paper (discounted, Blackwell, and average optimality) (Puterman, 2014), so we simply refer to them as *policies* and denote them as $\Pi = \mathcal{A}^{\mathcal{S}}$. A policy $\pi \in \Pi$ induces a vector of expected instantaneous reward $\boldsymbol{r}_\pi \in \mathbb{R}^{\mathcal{S}}$, defined as $r_{\pi,s} = r_{s\pi(s)}, \forall\, s \in \mathcal{S}$, as well as a Markov chain over $\mathcal{S}$, evolving via a transition matrix $\boldsymbol{P}_\pi \in \mathbb{R}^{\mathcal{S} \times \mathcal{S}}$, defined as $P_{\pi,ss'} = P_{s\pi(s)s'}, \forall\, s, s' \in \mathcal{S}$. We also write $r_\infty = \max\{|r_{sa}| \mid (s, a) \in \mathcal{S} \times \mathcal{A}\}$.

Given a discount factor $\gamma \in [0, 1)$ and a policy $\pi \in \Pi$, the *value function* $\boldsymbol{v}_\gamma^\pi \in \mathbb{R}^{\mathcal{S}}$ represents the discounted value obtained starting from each state: $v_{\gamma,s}^\pi = \mathbb{E}^{\pi,\boldsymbol{P}} \left[ \sum_{t=0}^{+\infty} \gamma^t r_{s_t, a_t} \mid s_0 = s \right], \forall\, s \in \mathcal{S}$. We start with discounted optimality, the most popular optimality criterion in RL.

**Definition 2.1.** Given $\gamma \in [0, 1)$, a policy $\pi \in \Pi$ is $\gamma$-discount-optimal if $v_{\gamma,s}^\pi \geq v_{\gamma,s}^{\pi'}, \forall\, \pi' \in \Pi, \forall\, s \in \mathcal{S}$. We call $\Pi_\gamma^\star \subset \Pi$ the set of $\gamma$-discount-optimal policies.

The discount factor $\gamma \in [0, 1)$ represents the preference for current rewards compared to future rewards. The difficulty of choosing the discount factor $\gamma$ is well recognized in RL (Tang et al., 2021). In some applications, it is reasonable to choose values of $\gamma$ close to 1, e.g., in finance (Deng et al., 2016), in healthcare (Garcia et al., 2021, Neumann et al., 2016) or in game solving (Brockman et al., 2016). In other applications, $\gamma$ is merely treated as a parameter introduced for algorithmic purposes, e.g., controlling the variance of the policy gradient estimates (Baxter and Bartlett, 2001), or ensuring convergence of algorithms. A discount-optimal policy can be computed efficiently with value iteration, policy iteration, and linear programming (Puterman, 2014). Notably, these algorithms do not require any assumptions on the MDP instance $\mathcal{M}$.

Another fundamental optimality criterion is *average optimality*, where the average reward $\boldsymbol{g}^\pi \in \mathbb{R}^{\mathcal{S}}$ of a policy $\pi \in \Pi$ is $g_s^\pi = \lim_{T \to +\infty} \frac{1}{T+1} \mathbb{E}^{\pi,\boldsymbol{P}} \left[ \sum_{t=0}^{T} r_{s_t, a_t} \mid s_0 = s \right], \forall\, s \in \mathcal{S}$. This limit always

exists for stationary policies (Puterman, 2014). A policy $\pi$ is average-optimal if $\boldsymbol{g}^\pi \geq \boldsymbol{g}^{\pi'}, \forall \, \pi' \in \Pi$. Average optimality has been extensively studied in the RL literature, as it alleviates the introduction of a potentially artificial discount factor. Classical algorithms include relative value iteration (Dong et al., 2019, Yang et al., 2016), and gradient-based methods (Bhatnagar et al., 2007, Iwaki and Asada, 2019). We refer to (Dewanto et al., 2020) for a survey on average optimality in RL.

Several technical complications arise from considering average optimality instead of discounted optimality. First, the average reward $\boldsymbol{g}^\pi$ of a policy is not a continuous function of the policy $\pi$ (e.g., chapter 4, (Feinberg and Shwartz, 2012)). This can make gradient-based methods inefficient, since a small change in the policy may result in drastic changes in the average reward. Additionally, the Bellman operator associated with the average optimality criterion is not a contraction and may have multiple fixed points. These complications can be circumvented by assuming structural properties on the MDP instance, such as bounded times of first returns and weakly-communicating MDPs (Akian and Gaubert, 2013, Wang et al., 2022). Some of these assumptions may be hard to verify in a simulation environment where only samples are available, or NP-hard to verify even when the MDP instance is fully known, as is the case for the unichain assumption (Tsitsiklis, 2007). One of our goals in this paper is to provide a method to compute average-optimal policies via solving discounted MDPs, *without any restrictive structural assumptions on the MDP instance*. We will do so via the notion of *Blackwell optimality*.

# 3 Classical theory of Blackwell optimality

In this section, we describe the classical definition of Blackwell optimality in MDPs and summarize its main limitations. We first give this definition of a Blackwell-optimal policy and outline the proof of its existence. This proof will serve as a building block of our main result in Section 4. We then highlight the main limitations of the existing definition of Blackwell optimality.

**Existing definition and algorithms.** We start with the following classical definition.

**Definition 3.1.** A policy $\pi$ is *Blackwell-optimal* if there exists $\gamma \in [0, 1)$, such that $\pi \in \Pi^\star_{\gamma'}, \ \forall \gamma' \in [\gamma, 1)$. We call $\Pi^\star_{\mathsf{bw}}$ the set of Blackwell-optimal policies.

In short, a Blackwell-optimal policy is $\gamma$-discount-optimal for all discount factors $\gamma$ sufficiently close to 1 (Blackwell, 1962). This notion has become popular in the field of reinforcement learning, mainly due to its connection to average optimality (Dewanto and Gallagher, 2021). Blackwell optimality bridges the gap between the different optimality criteria: it is defined in terms of discounted optimality, yet, crucially, Blackwell-optimal policies are average-optimal (theorem 10.1.5, (Puterman, 2014)). Therefore, any advances in computing Blackwell-optimal policies transfer to advances in computing average-optimal policies. A Blackwell-optimal policy is guaranteed to exist for finite MDPs.

**Theorem 3.2** ((Blackwell, 1962)). *When $|\mathcal{S}| < +\infty, |\mathcal{A}| < +\infty$, there exists at least one Blackwell-optimal policy: $\Pi^\star_{\mathsf{bw}} \neq \emptyset$.*

We highlight the proof of Theorem 3.2 based on section 10.1.1 in (Puterman, 2014). Summarizing this proof is important because it is not well-known and serves as a building block for our results.

*Step 1.* Let $\pi, \pi' \in \Pi, s \in \mathcal{S}$. Through this paper use the notation $\phi_s^{\pi,\pi'}$ for $\phi_s^{\pi,\pi'} : \gamma \mapsto v_{\gamma,s}^\pi - v_{\gamma,s}^{\pi'}$. We first show that $\phi_s^{\pi,\pi'}$ has finitely many zeros in $[0, 1)$. This is a consequence of the next lemma.

**Lemma 3.3.** *For $\pi \in \Pi$ and $s \in \mathcal{S}$, $\gamma \mapsto v_{\gamma,s}^\pi$ is a* rational function *on $[0, 1)$, i.e., it is the ratio of two polynomials.*

Lemma 3.3 follows from the Bellman equation for the value function $\boldsymbol{v}^\pi$: $\boldsymbol{v}^\pi = \boldsymbol{r}_\pi + \gamma \boldsymbol{P}_\pi \boldsymbol{v}^\pi$. Therefore, $\boldsymbol{v}^\pi$ is the unique solution to the equation $\boldsymbol{A}\boldsymbol{x} = \boldsymbol{b}$, for $\boldsymbol{b} = \boldsymbol{r}_\pi$ and $\boldsymbol{A} = \boldsymbol{I} - \gamma \boldsymbol{P}_\pi$. Lemma 3.3 then follows directly from Cramer's rule for the solution of a system of linear equations: since $\boldsymbol{A}$ is invertible, then $\boldsymbol{A}\boldsymbol{x} = \boldsymbol{b}$ has a unique solution $\boldsymbol{x}$, which satisfies $x_s = \det(\boldsymbol{A}_s)/\det(\boldsymbol{A}), \forall \, s \in \mathcal{S}$, with $\det(\cdot)$ the determinant of a matrix and $\boldsymbol{A}_s$ the matrix formed by replacing the $s$-th column of $\boldsymbol{A}$ by the vector $\boldsymbol{b}$. A consequence of Lemma 3.3 is that the function $\phi_s^{\pi,\pi'}$ is a rational function, and therefore its zeros are the zeros of a polynomial. This shows that $\phi_s^{\pi,\pi'}$ is either identically equal to 0, or it has only has finitely many roots in $[0, 1)$.

*Step 2.* We now conclude the proof of Theorem 3.2. Let $\pi, \pi' \in \Pi, s \in \mathcal{S}$. If $\phi_s^{\pi,\pi'}$ is not identically equal to 0, let $\gamma(\pi, \pi', s) \in [0, 1)$ be its the largest zero of $\phi_s^{\pi,\pi'}$ in $[0, 1)$: $\gamma(\pi, \pi', s) = \max\{\gamma \in [0, 1) | v_{\gamma,s}^\pi - v_{\gamma,s}^{\pi'} = 0\}$. We let $\gamma(\pi, \pi', s) = 0$ if $\phi_s^{\pi,\pi'}$ is identically equal to 0 in $[0, 1)$. We now let

$$\bar{\gamma} = \max_{\pi, \pi' \in \Pi, s \in \mathcal{S}} \gamma(\pi, \pi', s). \tag{3.1}$$

We have $\bar{\gamma} < 1$ since there is a finite number of (stationary, deterministic) policies and $|\mathcal{S}| < +\infty$. Let $\pi$ be $\gamma$-discount-optimal for a certain $\gamma > \bar{\gamma}$. We have, for any $s \in \mathcal{S}, v_{\gamma,s}^\pi \geq v_{\gamma,s}^{\pi'}, \forall \pi' \in \Pi$. By definition of $\bar{\gamma}$, the map $\phi_s^{\pi,\pi'}$ cannot change a sign on $[\bar{\gamma}, 1)$ (because it cannot be equal to 0), for any policy $\pi' \in \Pi$ and any state $s \in \mathcal{S}$, i.e., we have $v_{\gamma',s}^\pi \geq v_{\gamma',s}^{\pi'}, \forall \pi' \in \Pi, \forall \gamma' \in (\gamma, 1)$. This shows that $\pi$ remains $\gamma'$-discount-optimal for all $\gamma' > \gamma$, and, therefore, $\pi$ is Blackwell-optimal.

**Remark 3.4.** At this point, the reader may wonder if some Blackwell optimal policies are "better" than others, e.g., for instance, if we can find a Blackwell optimal policy that is $\gamma$-discount optimal for $\gamma$ as small as possible. Interestingly, all Blackwell optimal policies are $\gamma$-discount optimal (or not) for the same discount factors. This follows from the key property that the value functions of Blackwell optimal policies coincide for all $\gamma \in (0, 1)$ at all states $s \in \mathcal{S}$. Indeed, these value functions must coincide on an entire interval close enough to 1, and they are rational functions. Hence, if they are equal for an infinite number of discount factors, they are equal on the entire interval $(0, 1)$.

To the best of our knowledge, there are only two **existing algorithms** to compute a Blackwell-optimal policy. The first algorithm (Hordijk et al., 1985) formulates MDPs with varying discount factors as linear programs (LPs) over the field of power series with potentially negative coefficients, known as Laurent series. The simplex method for solving LPs over power series explores $[0, 1)$ and computes the subintervals of $[0, 1)$ where an optimal policy can be chosen constant (as a function of $\gamma$). It returns a Blackwell-optimal policy in a finite number of operations, but there are no complexity guarantees for this algorithm. The second algorithm is based on $n$-discount optimality, described with a set of $(|\mathcal{S}| + 1)$-nested equations indexed by $n = -1, ..., |\mathcal{S}| - 1$ that need to be solved sequentially by solving three LPs at each stage $n$ (O'Sullivan and Veinott Jr, 2017). This gives a polynomial-time algorithm for computing Blackwell-optimal policies, requiring solving $3(|\mathcal{S}| + 1)$ linear programs of dimension $O(|\mathcal{S}|)$. A simpler description is in section 10.3.4 in (Puterman, 2014), but only finite convergence is proved. We are not aware of any available implementations of these algorithms.

**Limitations of existing approaches.** We now highlight the shortcomings of the existing definition of Blackwell optimality. In particular, we demonstrate that the current approach is insufficient to reduce Blackwell optimality to discount optimality, we show that it does not lead to simple algorithms, and we show that it completely overlooks the potential pathological behaviors of the value functions.

First, Definition 3.1 leads to methods that are significantly more involved than solving discounted MDPs. The two existing algorithms for computing Blackwell-optimal policies handle complex objects, e.g., the simplex algorithm over the field of power series and nested optimality equations with multiple subproblems that need to be solved sequentially. The intricacy of both algorithms makes them difficult to implement, and these algorithms are not widely used in practice.

Second, Definition 3.1 implicitly introduces, for each Blackwell-optimal policy $\pi \in \Pi_{\text{bw}}^\star$, a discount factor $\gamma(\pi) \in [0, 1)$, defined as the smallest discount factor after which $\pi$ remains discount-optimal:

$$\gamma(\pi) = \min\{\gamma \in [0, 1) \mid \pi \in \Pi_{\gamma'}^\star, \forall \gamma' \in [\gamma, 1)\}. \tag{3.2}$$

We now show that $\gamma(\pi)$ provides insufficient information to compute a Blackwell-optimal policy.

**Proposition 3.5.** *There exists an MDP instance $\mathcal{M}$, a Blackwell-optimal policy $\pi \in \Pi_{\text{bw}}^\star$, and discount factors $\gamma_1, \gamma_2 \in [0, 1)$ with $\gamma_1 < \gamma(\pi) < \gamma_2$ such that:*

1. *the policy $\pi$ is $\gamma_1$-discount-optimal, and*
2. *there exists $\pi' \neq \pi$ that is $\gamma_2$-discount-optimal and* not *Blackwell-optimal.*

Proposition 3.5 shows the naive approach of solving a $\gamma$-discounted MDP for discount factor $\gamma > \gamma(\pi)$ does not compute a Blackwell-optimal policy. That is, the policy $\pi'$ in Proposition 3.5 is optimal for $\gamma_2 > \gamma(\pi)$ but is not Blackwell-optimal. It also shows that $\gamma(\pi)$ is not even the smallest discount factor for which $\pi$ is discount-optimal. Note that we are the first to highlight this shortcoming of the classical definition of Blackwell optimality. We also note that Proposition 3.5 remains true even under the assumption that MDP instance is unichain, as we prove in Appendix A.Overall, we have

shown that the discount factor $\gamma(\pi)$, appearing in the classical definition of Blackwell optimality, cannot be exploited to compute a Blackwell-optimal policy.

The limitation outlined above calls for the definition of another discount factor that can adequately describe when does the set of discount-optimal policies equals to the set of Blackwell optimal policies. We introduce this *Blackwell discount factor* in the next section. The proof of Proposition 3.5 is based on the following very simple example, with $|\mathcal{S}| = 8, |\mathcal{A}| = 3$, and deterministic transitions.

**Example 3.6.** *We consider the MDP instance from Figure 1. The decision maker starts in state* $0$ *and chooses one of three actions* $\{a_1, a_2, a_3\}$*; there is no choice in other states, all transitions are deterministic, and the rewards are indicated above the transition arcs. The reward for* $a_1$ *is 1 and the process transitions to the absorbing state* $7$*, which gives a reward of* $0$*. The reward for* $a_2$ *is 0, and the process transitions to states* $1, 2, 3$ *before reaching the absorbing state* $7$*. The value functions equal to* $v_\gamma^{a_2} = r_1\gamma + r_2\gamma^2$, $v_\gamma^{a_3} = r_4\gamma + r_5\gamma^2$, $v_\gamma^{a_1} = 1$*. Choosing* $(r_1, r_2) = (6, -8)$ *and* $(r_4, r_5) = (8/3, -16/9)$ *gives the value functions shown in Figure 1 (left figure). In particular,* $v_\gamma^{a_2}$ *is the parabola that is equal to* $0$ *at* $\gamma = 0$*, and equal to 1 at* $\gamma \in \{1/4, 1/2\}$*, and* $v_\gamma^{a_3}$ *is the parabola that is equal to* $0$ *at* $\gamma = 0$ *and equal to its maximum 1 at* $\gamma = 3/4$*. This shows that* $a_1$ *is Blackwell-optimal with* $\gamma(a_1) = 1/2$*. Additionally, for* $\gamma_1 \in [0, 1/4]$*,* $a_1$ *is* $\gamma_1$*-discount-optimal. Finally,* $a_3$ *is* $\gamma_2$*-discount-optimal for* $\gamma_2 = 3/4$*, but it is not Blackwell-optimal.*

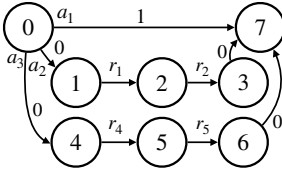
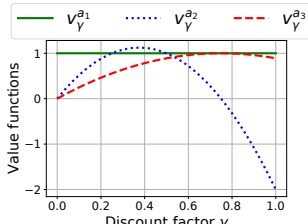

Figure 1: MDP instance (left) and value functions (right) for Example 3.6.

In the next proposition, we show that the subintervals of $[0, 1)$ where a policy is discount-optimal may be much more complex than usually alluded to in the literature. In particular, there exists a simple MDP instance with only two policies, but where a Blackwell-optimal policy may be discount-optimal in an *arbitrary* number of *arbitrary* disjoint subintervals of $[0, 1)$.

**Theorem 3.7.** *For any odd integer* $N \in \mathbb{N}$ *and any sequence* $0 = \gamma_0 < \gamma_1 < ... < \gamma_{N-1} < \gamma_N = 1$*, there exists an MDP instance* $(\mathcal{S}, \mathcal{A}, \boldsymbol{r}, \boldsymbol{P})$ *with* $|\mathcal{S}| = N + 1$ *and* $|\mathcal{A}| = 2$*, and two policies* $\pi_1, \pi_2$ *such that* $\pi_1$ *is the unique optimal policy on any of the intervals* $(\gamma_{2i}, \gamma_{2i+1})$ *for* $i = 0, ..., (N-1)/2$ *and* $\pi_2$ *is the unique optimal policy on* $(\gamma_{2i-1}, \gamma_{2i})$*, for* $i = 1, ..., (N-1)/2$*.*

Theorem 3.7 shows that the algorithm that explore the entire interval of $(0, 1)$ to compute discount-optimal policies (Hordijk et al., 1985) may visit a number of subintervals that is impractical. We present a detailed proof in Appendix B. The proof relies on interpreting value functions as polynomials and using Lagrange interpolation polynomials to tune the instantaneous rewards to ensure that the value functions intersect at the given discount factors. Overall, our results in this section highlight the pitfalls of the existing approach to Blackwell optimality and the potential pathological behaviors of the value functions, even in simple MDP instances. We ameliorate this issue in the next section.

## 4 Introducing the Blackwell discount factor

This section introduces the notion of the *Blackwell discount factor*, which we use to reduce Blackwell optimality and average optimality to discounted optimality. This reduction leads to algorithms to compute Blackwell-optimal and average policies that are significantly simpler than the state-of-the-art. Intuitively, we need the following condition to reduce Blackwell optimality to discounted optimality: there must exist a discount factor $\gamma_{\mathsf{bw}} \in [0, 1)$ such that any $\gamma$-discount-optimal policy for $\gamma > \gamma_{\mathsf{bw}}$ is also $\gamma'$-discount-optimal for any other $\gamma' > \gamma_{\mathsf{bw}}$. The following definition formalizes this intuition.

**Definition 4.1.** The Blackwell discount factor $\gamma_{\mathsf{bw}} \in [0, 1)$ is equal to $\gamma_{\mathsf{bw}} = \inf\{\gamma \in [0, 1) \mid \Pi_{\gamma'}^\star = \Pi_{\mathsf{bw}}^\star, \forall \ \gamma' \in (\gamma, 1)\}$, where $\Pi_{\mathsf{bw}}^\star$ is the set of Blackwell-optimal policies.

We establish the existence of a Blackwell discount factor in the next theorem.

**Theorem 4.2.** *The Blackwell discount factor $\gamma_{\mathsf{bw}}$ in Definition 4.1 exists in any finite MDP.*

*Proof.* We show that there exists a discount factor $\gamma \in [0, 1)$ such that $\Pi_{\gamma'}^\star = \Pi_{\mathsf{bw}}^\star, \forall \ \gamma' \in (\gamma, 1)$. Let $\bar{\gamma}$ defined as in Equation (3.1). We show $\forall \ \gamma \in [\bar{\gamma}, 1), \Pi_\gamma^\star = \Pi_{\mathsf{bw}}^\star$. Let $\gamma' \in (\bar{\gamma}, 1)$ and let $\pi$ be a policy that is $\gamma'$-discount-optimal. By definition, we have $v_{\gamma',s}^\pi \geq v_{\gamma',s}^{\pi'}, \forall \ \pi' \in \Pi, \forall \ s \in \mathcal{S}$. Since $\gamma' > \bar{\gamma}$, the map $\phi_s^{\pi,\pi'}$ does not change sign on $[\bar{\gamma}, 1)$. This shows that $\pi$ is $\gamma$-discount-optimal for all $\gamma \in (\bar{\gamma}, 1)$. Therefore, $\pi$ is Blackwell optimal, and any $\gamma$-discount-optimal policy is Blackwell optimal, for any $\gamma \in (\bar{\gamma}, 1)$, i.e., this shows $\Pi_{\bar{\gamma}}^\star \subset \Pi_{\mathsf{bw}}^\star$. The inclusion $\Pi_{\mathsf{bw}}^\star \subset \Pi_{\bar{\gamma}}^\star$ follows from the definition of $\bar{\gamma}$: if $\pi$ is Blackwell-optimal but not discount-optimal for $\bar{\gamma}$, then it must become discount-optimal for a larger $\gamma' > \bar{\gamma}$, which is impossible since $\bar{\gamma}$ is the largest discount factors where the value functions of any two stationary policies can intersect. $\square$

**Difference from the existing definition.** It is important to elaborate on the difference between Definition 3.1 (classical definition of Blackwell optimality) and Definition 4.1 (Blackwell discount factor). While the proof for the existence of $\gamma_{\mathsf{bw}}$ is relatively concise, the distinction between $\gamma_{\mathsf{bw}}$ and $\gamma(\pi)$ has been utterly overlooked in the literature, where it is common to find statements that suggest that $\gamma > \gamma(\pi)$ implies Blackwell optimality of all discount-optimal policies, e.g. in Dewanto and Gallagher (2021), Wang et al. (2023). To the best of our knowledge, we are the first to properly introduce the Blackwell discount factor $\gamma_{\mathsf{bw}}$, to show its sufficiency to compute Blackwell-optimal policies, to emphasize the shortcomings of the classical approach to Blackwell optimality, and to clarify the distinction between $\gamma_{\mathsf{bw}}$ and $\gamma(\pi)$. In particular, in Definition 3.1, a Blackwell-optimal policy $\pi$ is optimal for any $\gamma \in [\gamma(\pi), 1)$. However, for some $\gamma \in [\gamma(\pi), 1)$, there may be other optimal policies that are not Blackwell-optimal, as shown in Proposition 3.5. We show an MDP instance like this in Example 3.6, where $\gamma_{\mathsf{bw}} = 3/4$ but where $\gamma(a_1) = 1/2$, and $a_1$ is the only Blackwell-optimal policy. Hence in all generality, we may have $\gamma(\pi) < \gamma_{\mathsf{bw}}$, and $\gamma(\pi) \neq \gamma_{\mathsf{bw}}$. Note that the authors in (Dewanto and Gallagher, 2021, Dewanto et al., 2020) also introduce the notation "$\gamma_{\mathsf{bw}}$" but they use it to denote $\gamma(\pi)$.

**Reduction to discounted optimality.** If $\gamma_{\mathsf{bw}}$ is known for a given MDP instance, it is straightforward to compute a Blackwell-optimal policy, by solving a discounted MDP with $\gamma > \gamma_{\mathsf{bw}}$. Therefore, the notion of Blackwell discount factor provides a method to reduce the criteria of Blackwell optimality and average optimality to the well-studied criterion of discounted optimality. As we have discussed before, efficient methods for solving discounted MDPs such as value iteration or linear programming have been extensively studied. These algorithms are much simpler than the two existing algorithms for computing Blackwell-optimal policies. Note that it is enough to compute an upper bound on $\gamma_{\mathsf{bw}}$. In particular, if we are able to show that $\gamma_{\mathsf{bw}} < \gamma'$ for some $\gamma' \in [0, 1)$, then following the definition of $\gamma_{\mathsf{bw}}$, we can compute a Blackwell-optimal policy by solving a discounted MDP with a discount factor $\gamma = \gamma'$. Therefore, in the rest of Section 4, we focus on obtaining an upper bound on $\gamma_{\mathsf{bw}}$.

## 4.1 Upper bound on $\gamma_{\mathsf{bw}}$

We now obtain an instance-dependent upper bound on $\gamma_{\mathsf{bw}}$, i.e., we construct a scalar $\eta(\mathcal{M}) \in (0, 1)$ for each MDP instance $\mathcal{M} = (\mathcal{S}, \mathcal{A}, \boldsymbol{r}, \boldsymbol{P})$, such that $\gamma_{\mathsf{bw}} < 1 - \eta(\mathcal{M})$. Our main contribution in this section is Theorem 4.4, which gives a closed-form expression for $\eta(\mathcal{M})$ as a function of the *maximum bit-size* of the data of the MDP instance $\mathcal{M}$. We start by showing that it is impossible to obtain a bound on $\gamma_{\mathsf{bw}}$ that is independent of $\boldsymbol{r}$ or $\boldsymbol{P}$.

**Proposition 4.3.** *For any $\eta > 0$, there exists an MDP instance $\mathcal{M} = (\mathcal{S}, \mathcal{A}, \boldsymbol{r}, \boldsymbol{P})$ with $|\mathcal{S}| = 2, |\mathcal{A}| = 2$ and deterministic transitions, such that $\gamma_{\mathsf{bw}} > 1 - \eta$.*

*Proof.* Let $\mathcal{S} = \{s_1, s_2\}, \mathcal{A} = \{a_1, a_2\}$. In state $s_1$, action $a_1$ transitions to $s_1$ (with reward 0) and action $a_2$ transitions to $s_2$ (with reward $-1$). There is no action to choose in state $s_2$ which is absorbing with a reward $\epsilon > 0$. It is straightforward to check that $a_2$ is Blackwell optimal, with $\gamma_{\mathsf{bw}} = (1 + \epsilon)^{-1}$, so that $\gamma_{\mathsf{bw}}$ can be chosen arbitrarily close to 1 by choosing small values for $\epsilon$. $\square$

We show that Proposition 4.3 still holds even under the assumption that the MDP instances are weakly-communicating in Appendix C. Proposition 4.3 shows that an instance-dependent bound on $\gamma_{\mathsf{bw}}$ *must* depend on the "coarseness" of $r$ and $P$. This suggests parametrizing our upper bound by the *bit-sizes* of the MDP instance. MDPs with finite bit-sizes parameters are the MDP instances that can be exactly encoded in a computer and practically solved by existing algorithms. We first recall the definitions pertaining to bit-size, necessary to describe the complexity of classical weakly-polynomial time algorithms like interior-point methods (section 4.6 in (Ben-Tal and Nemirovski, 2001)) and the ellipsoid method (Bland et al., 1981). The bit-size of $r \in \mathbb{N}$ is $\lfloor \log_2(r) \rfloor$, the number of bits necessary to represent $r$ with standard binary encoding. The bit-size of a rational number is the sum of the bit-size of its numerator and its denominator. The maximum bit-size of an MDP instance is the maximum bit-size of any $r_{sa}$ and $P_{sas'}$ for $(s, a, s') \in \mathcal{S} \times \mathcal{A} \times \mathcal{S}$. Its total bit-size is the sum of the bit-sizes of the components of $r$ and $P$. For instance, in the riverswim instance, the maximum bit-size of the reward is $14$, since the largest rewards are bounded by $10^4$ in the terminal states. Our main theorem in this section provides a strict upper bound on $\gamma_{\mathsf{bw}}$ as follows.

**Theorem 4.4.** *Let $\mathcal{M} = (\mathcal{S}, \mathcal{A}, r, P)$ be an MDP instance with finite bit-size and let $m \in \mathbb{N}$ be the maximum bit-size of the instance $\mathcal{M}$. Then we have $\gamma_{\mathsf{bw}} < 1 - \eta(\mathcal{M})$, with $\eta(\mathcal{M}) \in (0, 1)$ defined as*

$$\eta(\mathcal{M}) = \frac{1}{2 N^{N/2+2} (L+1)^N}, N = 2|\mathcal{S}| - 1, L = 2 \cdot |\mathcal{S}| \cdot r_\infty \cdot m^{2|\mathcal{S}|} \cdot 4^{|\mathcal{S}|}.$$

Our proof uses ideas that are new in the MDP literature, such as the separation of algebraic numbers. We provide an outline of the proof below and defer the full statement to Appendix D.

In the first step of the proof, by carefully inspecting the proofs of Theorem 3.2 and of Theorem 4.2, we note that an upper bound for $\gamma_{\mathsf{bw}}$ is $\bar{\gamma}$, as defined in (3.1): $\bar{\gamma} = \max_{\pi,\pi' \in \Pi, s \in \mathcal{S}} \gamma(\pi, \pi', s)$, where for $\pi, \pi' \in \Pi$ and $s \in \mathcal{S}$, $\gamma(\pi, \pi', s)$ is the largest discount factor $\gamma$ in $[0, 1)$ for which $\phi_s^{\pi,\pi'}(\gamma) = 0$ when $\phi_s^{\pi,\pi'} : \gamma \mapsto v_{\gamma,s}^\pi - v_{\gamma,s}^{\pi'}$ is not identically equal to $0$, and $0$ otherwise. Therefore, we focus on obtaining an upper bound on $\gamma(\pi, \pi', s)$ for any two policies $\pi, \pi' \in \Pi$ and any state $s \in \mathcal{S}$.

In the second step, following Lemma 3.3, the value functions $\gamma \mapsto v_s^\pi, \gamma \mapsto v_s^{\pi'}$ are rational functions, i.e., they are ratios of two polynomials. Therefore, we interpret $\phi_s^{\pi,\pi'}(\gamma) = 0$ as a polynomial equation in $\gamma$, i.e., as $p(\gamma) = 0$ for a certain polynomial $p$. With this notation, $\gamma(\pi, \pi', s) \in [0, 1)$ is a root of $p$. We show that $\gamma = 1$ is always a root of $p$, even though value functions a priori not defined for $\gamma = 1$. We then precisely characterize the degree $N$ and the sum $L$ of the absolute values of the coefficients of the polynomial $p$, depending on the MDP instance $\mathcal{M}$.

**Theorem 4.5.** *The polynomial $p$ has degree $N = 2|\mathcal{S}| - 1$. Moreover, $m^{2|\mathcal{S}|}p$ has integral coefficients. The sum of the absolute values of the coefficients of $m^{2|\mathcal{S}|}p$ is bounded by $L = 2 \cdot |\mathcal{S}| \cdot r_\infty \cdot m^{2|\mathcal{S}|} \cdot 4^{|\mathcal{S}|}$.*

In the third step, we lower-bound the distance between any two distinct roots of $p$. To do this, we rely on the following *separation bounds of algebraic numbers*.

**Theorem 4.6** ((Rump, 1979)). *Let $p$ be a polynomial of degree $N$ with integer coefficients. Let $L$ be the sum of the absolute values of its coefficients. The distance between any two distinct roots of $p$ is strictly larger than $\eta > 0$, with $\eta = 2N^{-N/2+2}(L+1)^{-N}$.*

Recall that $\gamma(\pi, \pi', s)$ and $1$ are two always roots of $p$, with $\gamma(\pi, \pi', s) < 1$. Combining Theorem 4.5 with Theorem 4.6, we conclude that $\gamma(\pi, \pi', s) < 1 - \eta(\mathcal{M})$ for $\eta(\mathcal{M}) > 0$ defined as in Theorem 4.4. Therefore, $\bar{\gamma} < 1 - \eta(\mathcal{M})$, and $\gamma_{\mathsf{bw}} < 1 - \eta(\mathcal{M})$. This concludes our proof of Theorem 4.4.

**Discussion.** Using Theorem 4.4, we obtain the first reduction from Blackwell optimality to discounted optimality: solving a discounted MDP with $\gamma \geq 1 - \eta(\mathcal{M})$ returns a Blackwell-optimal policy. Blackwell optimality implies average optimality, so we also obtain the first reduction from average optimality to discounted optimality *without any assumptions on the structure of the underlying Markov chains of the MDP*. We also discuss the **complexity results** for computing a Blackwell-optimal policy using our reduction. Policy iteration returns a discounted optimal policy in $O\left(\frac{|\mathcal{S}|^2|\mathcal{A}|}{1-\gamma} \log\left(\frac{1}{1-\gamma}\right)\right)$ iterations (Scherrer, 2013), but it may be slow to converge when $\gamma = 1 - \eta(\mathcal{M})$ as in Theorem 4.4, since $\eta(\mathcal{M})$ may be close to $0$. Various algorithms exist to obtain convergence faster than $O(1/(1-\gamma))$, such as accelerated value iteration (Goyal and Grand-Clément, 2023a) and Anderson acceleration (Zhang et al., 2020). However, note that $\lfloor \log_2(\eta(\mathcal{M})) \rfloor$, the bit-size of the scalar $\eta(\mathcal{M})$,

is polynomial in the bit-size of the MDP instance $\mathcal{M}$. Since discounted MDPs can be formulated as linear programs, which can be solved in polynomial-time in the input size of the MDP (Ye, 2011), we obtain a weakly-polynomial time algorithm for computing Blackwell-optimal policies. We present the proof of the following theorem in Appendix E.

**Theorem 4.7.** *Let $\mathcal{M} = (\mathcal{S}, \mathcal{A}, \boldsymbol{r}, \boldsymbol{P})$ be an MDP instance with total bit-size $Q(\boldsymbol{r}, \boldsymbol{P}) \in \mathbb{N}$. Then we can compute a Blackwell-optimal policy in $O\left(|\mathcal{S}|^5 |\mathcal{A}|^2 Q(\boldsymbol{r}, \boldsymbol{P})\right)$ arithmetic operations.*

Note that with Theorem 4.4 and Theorem 4.7, we have reduced the complex problem of computing a Blackwell optimal policy to a much simpler and well-studied problem: solving a linear program, which can be done in weakly-polynomial time. Potential improvements for our upper bound on $\gamma_{\mathsf{bw}}$ are an important future direction: more precise separation bounds than Theorem 4.6 could be obtained for the specific polynomial $p$ appearing in the proof of Theorem 4.4, or for a specific MDP instances, e.g. ergodic or unichain MDPs. Going beyond the case of finite sets of states and actions is interesting but this may be difficult, as in both cases there may not exist a Blackwell optimal policy anymore (Chitashvili, 1976, Maitra, 1965).

## 4.2 The case of robust MDPs

In practice, the value function $\boldsymbol{v}_\gamma^\pi$ may be very sensitive to the values of the transition probabilities $\boldsymbol{P}$. To emphasize this dependence, in this section we note $\boldsymbol{v}_\gamma^{\pi, \boldsymbol{P}}$ for the value function associated with a policy $\pi$ and a transition probability $\boldsymbol{P}$, defined similarly as in Section 2. Robust MDPs (RMDPs) ameliorate this issue by considering an *uncertainty set* $\mathcal{U}$, which can be seen as a plausible region for the transition probabilities $\boldsymbol{P} \in \mathcal{U}$. We focus on the case of sa-rectangular MDPs (Iyengar, 2005), where $\mathcal{U} = \times_{(s,a) \in \mathcal{S} \times \mathcal{A}} \mathcal{U}_{sa}$ for $\mathcal{U}_{sa} \subseteq \Delta(\mathcal{S})$. The worst-case value function $\boldsymbol{v}_\gamma^{\pi, \mathcal{U}} \in \mathbb{R}^{\mathcal{S}}$ of a policy $\pi$ is defined as $v_{\gamma, s}^{\pi, \mathcal{U}} = \min_{\boldsymbol{P} \in \mathcal{U}} v_{\gamma, s}^{\pi, \boldsymbol{P}}, \forall\, s \in \mathcal{S}$. In discounted RMDPs, the goal is to compute a *robust discounted optimal* policy, defined as follows.

**Definition 4.8.** Given $\gamma \in [0, 1)$, a policy $\pi \in \Pi$ is robust $\gamma$-discount-optimal if $v_{\gamma, s}^{\pi, \mathcal{U}} \geq v_{\gamma, s}^{\pi', \mathcal{U}}, \forall\, \pi' \in \Pi, \forall\, s \in \mathcal{S}$. We write $\Pi_{\gamma, \mathsf{rob}}^\star$ the set of robust $\gamma$-discount-optimal policies.

Robust Blackwell optimality is studied in (Goyal and Grand-Clément, 2023b, Tewari and Bartlett, 2007), to address the sensitivity of the robust value functions as regards the discount factors. Its connection to average reward RMDPs is discussed in (Tewari and Bartlett, 2007, Wang et al., 2023).

**Definition 4.9.** A policy $\pi \in \Pi$ is *robust Blackwell-optimal* if there exists $\gamma \in [0, 1)$, such that $\pi \in \Pi_{\gamma', \mathsf{r}}^\star, \forall\, \gamma' \in [\gamma, 1)$. We call $\Pi_{\mathsf{bw}, \mathsf{r}}^\star$ the set of robust Blackwell-optimal policies.

(Goyal and Grand-Clément, 2023b) shows the existence of a Blackwell-optimal policy for RMDPs, under the condition that $\mathcal{U}$ is sa-rectangular and has finitely many extreme points. This is the case for popular polyhedral uncertainty sets, e.g., when $\mathcal{U}_{sa}$ is based on the $\ell_p$ distance, for $p \in \{1, \infty\}$ (Givan et al., 1997, Ho et al., 2018, Iyengar, 2005), for some estimated kernel $\boldsymbol{P}^0$ and some radius $\alpha_{sa} > 0$:

$$\mathcal{U}_{sa} = \{\boldsymbol{p} \in \Delta(\mathcal{S}) \mid \|\boldsymbol{p} - \boldsymbol{P}_{sa}^0\|_p \leq \alpha_{sa}\}. \tag{4.1}$$

**Definition 4.10.** We define the robust Blackwell discount factor $\gamma_{\mathsf{bw}, \mathsf{r}} \in [0, 1)$ as $\gamma_{\mathsf{bw}, \mathsf{r}} = \inf\{\gamma \in [0, 1) \mid \Pi_{\gamma', \mathsf{r}}^\star = \Pi_{\mathsf{bw}, \mathsf{r}}^\star, \forall \gamma' \in (\gamma, 1)\}$.

We provide a detailed proof of the existence of the robust Blackwell discount factor in Appendix F. The proof strategy is the same as for the existence of the Blackwell discount factor for MDPs. We can obtain the same upper bound on $\gamma_{\mathsf{bw}, \mathsf{r}}$, by studying the values of $\gamma$ for which $\gamma \mapsto v_{\gamma, s}^{\pi, \boldsymbol{P}} - v_{\gamma, s}^{\pi', \boldsymbol{P}'}$ cancels, for any two policies $\pi, \pi' \in \Pi$ and any two extreme points $\boldsymbol{P}, \boldsymbol{P}'$ of $\mathcal{U}$. Writing $\gamma(\pi, \pi', s, \boldsymbol{P}, \boldsymbol{P}')$ for the largest zero in $[0, 1)$ of the function $\gamma \mapsto v_{\gamma, s}^{\pi, \boldsymbol{P}} - v_{\gamma, s}^{\pi', \boldsymbol{P}'}$ if it is not identically equal to zero, or $\gamma(\pi, \pi', s, \boldsymbol{P}, \boldsymbol{P}') = 0$ otherwise, an upper bound on $\gamma_{\mathsf{bw}, \mathsf{r}}$ for RMDPs can be computed as $\bar{\gamma}_\mathsf{r}$, defined as $\bar{\gamma}_\mathsf{r} = \max_{\pi, \pi' \in \Pi, s \in \mathcal{S}} \max_{\boldsymbol{P}, \boldsymbol{P}' \in \mathcal{U}_{\mathsf{ext}}} \gamma(\pi, \pi', s, \boldsymbol{P}, \boldsymbol{P}')$ with $\mathcal{U}_{\mathsf{ext}}$ the set of extreme points of $\mathcal{U}$. This leads to the following theorem.

**Theorem 4.11.** *Let $\mathcal{M} = \left(\mathcal{S}, \mathcal{A}, \boldsymbol{r}, \boldsymbol{P}^0\right)$ be an MDP instance with maximum bit-size $m \in \mathbb{N}$. Assume that $\mathcal{U}$ is sa-rectangular, where for each $(s, a) \in \mathcal{S} \times \mathcal{A}$, $\mathcal{U}_{sa}$ is constructed as in (4.1) based on $\ell_1$ or $\ell_\infty$ distance, and with the scalars $(\alpha_{sa})_{s, a}$ of maximum bit-size $m$. Then $\gamma_{\mathsf{bw}, \mathsf{r}} \leq 1 - \eta(\mathcal{M})$, with $\eta(\mathcal{M})$ defined as in Theorem 4.4 with $m' = 2m$ instead of $m$.*

Based on Theorem 4.11, we obtain the first reduction from robust Blackwell optimality to robust discounted optimality. Since discounted RMDPs can be solved with value iteration or policy iteration, we provide the first algorithms to compute a robust Blackwell-optimal policy for RMDPs with sa-rectangular uncertainty, when the uncertainty set is based on the $\ell_1$ or the $\ell_\infty$ distance. Note that there is no known convex (or linear) formulation for RMDPs (Grand-Clément and Petrik, 2022), so we are not able to provide a complexity statement akin to Theorem 4.7.

# 5 Conclusion

We highlight the shortcomings of the existing approach to Blackwell optimality and we introduce the Blackwell discount factor to ameliorate this issue. We provide an upper bound for MDPs and RMDPs in all generality, parametrized by the bit-sizes of the instances. Any progress in solving discounted MDPs, one of the most active research directions in RL, can be combined with our results to obtain new algorithms for computing average- and Blackwell-optimal policies. Our work also opens new research avenues for MDPs and RMDPs: the proof techniques for our bound on $\gamma_{\mathsf{bw}}$ and $\gamma_{\mathsf{bw,r}}$, based on the separation of algebraic numbers, are novel and they could be tightened for specific instances or different optimality criteria, such as bias optimality or $n$-discount optimality. The notion of *approximate* Blackwell optimality as well as the existence of the robust Blackwell discount factor for other uncertainty sets, e.g., s-rectangular or non-polyhedral sa-rectangular uncertainty sets, or for distributionally robust MDPs, are also interesting directions of research.

**Funding.** J. Grand-Clément is supported by the Agence Nationale de la Recherche [Grant 11-LABX-0047] and by Hi! Paris. M. Petrik's work was supported, in part, by NSF grants 2144601 and 1815275.

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

# A  Unichain instance for Proposition 3.5

We can extend Example 3.5 to a unichain MDP as follows: we add a transition from state 7 to state 0, with a reward of 0. We also add three intermediate states from 0 to 7 for action $a_1$, so that it takes as many periods to reach state 7 from state 0 for the three actions $a_1, a_2, a_3$. Note that this new MDP is unichain. We represent it in Figure 3a. Additionally, for this new MDP instance, we have $v_\gamma^{a_1} = 1/(1-\gamma^5), v_\gamma^{a_2} = (r_1\gamma + r_2\gamma^2)/(1-\gamma^5), v_\gamma^{a_3} = (r_4\gamma + r_5\gamma^2)/(1-\gamma^5)$, which are the same expressions as in Example 3.5, up to the common denominator $(1-\gamma^5)^{-1}$. Therefore, we have proved that the same conclusion as Proposition 3.4 holds for unichain MDPs.

# B  Proof of Theorem 3.7

*Proof.* Consider the following MDP instance, represented in Figure 2a. The initial state is state 0, where there are two actions to be chosen, $a_1$ or $a_2$. Action $a_1$ yields an instantaneous reward of 1 and then the decision maker transitions to the absorbing state $N$, where there is a reward of 0. Otherwise, choosing action $a_2$ yields an instantaneous reward $r_0$ and takes the decision maker through a deterministic sequence of states $1, ..., N-1$ with rewards $r_1, ..., r_{N-1}$, before transitioning to state $N$. For a given $\gamma \in [0, 1)$, the closed-form expressions for the value functions $v_\gamma^{a_1}, v_\gamma^{a_2}$ are $v_\gamma^{a_1} = 1$ and $v_\gamma^{a_2} = \sum_{t=0}^{N-1} r_t\gamma^t$.

Note that $\gamma \mapsto v_\gamma^{a_2}$ is a polynomial of degree $N-1$. Using Lagrange interpolation polynomials (section 0.9.11, (Horn and Johnson, 2012)), we can find coefficients $r_0, ..., r_{N-1}$ such that $\gamma \mapsto v_\gamma^{a_1}$ is equal to 1 for all $N-1$ discount factors $\gamma_1, ..., \gamma_{N-1}$ and equal to 0.9 at $\gamma_0 = 0$. The value function $v_\gamma^{a_2}$ resulting from this construction is highlighted in Figure 2b for $N = 5$ and $(\gamma_0, \gamma_1, \gamma_2, \gamma_3, \gamma_4, \gamma_5) = (0, 0.2, 0.4, 0.6, 0.8, 1.0)$. Let us note $q: \gamma \mapsto v_\gamma^{a_1} - v_\gamma^{a_2}$. Our choice of the rewards ensures that $q$ is a polynomial of degree $N-1$, with $q(0) > 0$, and $q(\gamma) = 0$ for $\gamma \in \{\gamma_1, ..., \gamma_{N-1}\}$. Because $\gamma \mapsto q(\gamma) - 1$ is a polynomial of degree $N-1$ with $N-1$ different real roots, it changes signs at every root. This shows that $\gamma \mapsto v_\gamma^{a_1} - v_\gamma^{a_2}$ is positive on $(\gamma_0, \gamma_1)$, negative on $(\gamma_1, \gamma_2)$, then positive on $(\gamma_2, \gamma_3)$, etc.. Action $a_1$ is optimal on $(\gamma_{N-1}, \gamma_N) = (\gamma_{N-1}, 1)$ because $N$ is odd. This concludes the proof of Theorem 3.7.

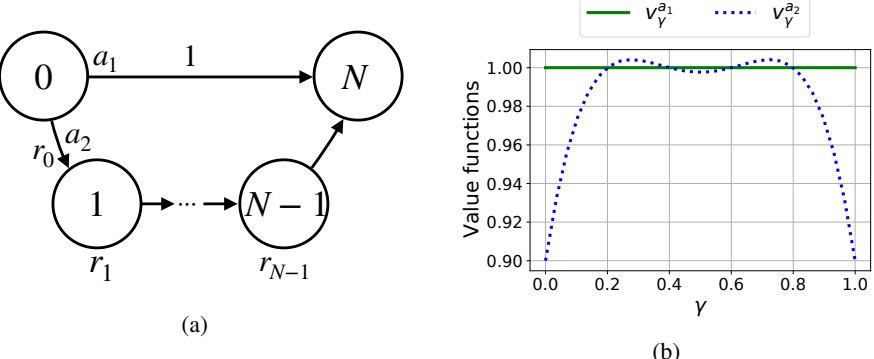

(a)

(b)

Figure 2: MDP instance for our proof of Theorem 3.7 (Figure 2a) and the value functions for $N = 5$ (Figure 2b).

$\square$

# C  Weakly-communicating instances for Proposition 4.3

Consider the MDP instance from the proof of Proposition 4.3. We now add a deterministic transition from state $s_2$ to state $s_1$, with a reward of 0 for action $a_1$ and a reward of $\epsilon$ for action $a_2$. The new MDP instance is represented in Figure 3b.

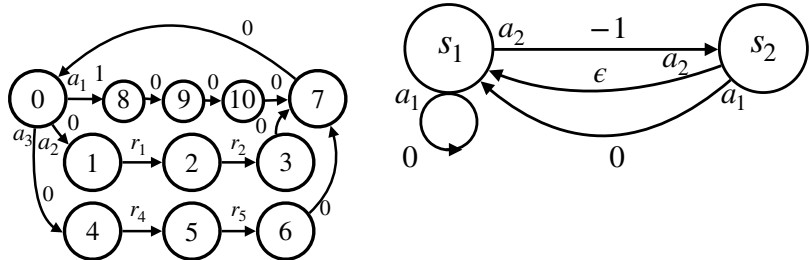

(a) Unichain instance for Proposition 3.5

(b) Weakly-communicating instance for Proposition 4.3

Figure 3: MDP instances to generalize Proposition 4.3 and Proposition 3.5.

First, this MDP instance is weakly-communicating since $\{s_1, s_2\}$ is strongly connected under policy $a_2$. In this new MDP instance, we still have $v_\gamma^{a_1} = 0$ but $v_\gamma^{a_2} = (-1 + \epsilon\gamma)/(1 - \gamma)$. Hence $a_2$ is Blackwell optimal when $\gamma \geq 1/\epsilon$. By choosing $\epsilon$ larger than 1 and $\epsilon \to 1$, we obtain $\gamma_{\mathsf{bw}} \to 1$. This shows that we can extend Proposition 4.3 to weakly-communicating MDPs.

# D   Proof of Theorem 4.4

In this appendix, we provide the proof for Theorem 4.4. As noted in Section 4.1, to bound $\gamma_{\mathsf{bw}}$, it is enough to obtain an upper bound on $\gamma(\pi, \pi', s)$ for any $\pi, \pi' \in \Pi$ and $s \in \mathcal{S}$ such that $\gamma \mapsto v_{\gamma,s}^\pi - v_{\gamma,s}^{\pi'}$ is not identically equal to 0, since $\gamma_{\mathsf{bw}} \leq \max_{\pi,\pi' \in \Pi, s \in \mathcal{S}} \gamma(\pi, \pi', s)$. Since $m$ is the maximum bit-size of the input data, we can write, for any $(s, a, s') \in \mathcal{S} \times \mathcal{A} \times \mathcal{S}$, $P_{sas'} = n_{sas'}/m$, for $n_{sas'} \in \mathbb{N}, n_{sas'} \leq m$, and $r_{sa} = q_{sa}/m, |q_{sa}| \leq r_\infty$. Examples of MDPs with finite bit-sizes include any real instances used for applications where the transition probabilities are estimated as empirical frequencies from some data, e.g. examining patients' transfers in hospitals as in (Hu et al., 2018) and (Grand-Clément et al., 2022), MDPs for hypertension treatment (Garcia et al., 2021), diabetes management (Steimle et al., 2021) and cancer detection (Goh et al., 2018), as well as the machine maintenance studied in (Wiesemann et al., 2013) and (Delage and Mannor, 2010). We now proceed to proving Theorem 4.4.

**Step 1.**   We start by studying in more detail the properties of the value functions. The following lemma follows directly from Cramer's rule, as explained in Section 3.

**Lemma D.1.** *We have*

$$v_{\gamma,s}^\pi = \frac{\det(\boldsymbol{M}(\gamma, s, \pi))}{\det(\boldsymbol{I} - \gamma\boldsymbol{P}_\pi)}, \tag{D.1}$$

*with $\boldsymbol{M}(\gamma, s, \pi)$ the matrix formed by replacing the $s$-th column of $\boldsymbol{I} - \gamma\boldsymbol{P}_\pi$ by the vector $\boldsymbol{r}_\pi$.*

From Lemma D.1, we have

$$v_{\gamma,s}^\pi = \frac{n(\gamma, s, \pi)}{d(\gamma, \pi)}$$

for $n(\gamma, s, \pi) = \det(\boldsymbol{M}(\gamma, s, \pi))$ and $d(\gamma, \pi) = \det(\boldsymbol{I} - \gamma\boldsymbol{P}_\pi)$. We choose the letter $n$ for *nominator* and the letter $d$ for *denominator*.

Note that $\gamma \mapsto n(\gamma, s, \pi)$ is a polynomial of degree at most $|\mathcal{S}|-1$, while $\gamma \mapsto d(\gamma, \pi)$ is a polynomial of degree at most $|\mathcal{S}|$.

We have, by definition,

$$
\begin{aligned}
v_{\gamma,s}^\pi - v_{\gamma,s}^{\pi'} &= \frac{n(\gamma, s, \pi)}{d(\gamma, \pi)} - \frac{n(\gamma, s, \pi')}{d(\gamma, \pi')} \\
&= \frac{n(\gamma, s, \pi)d(\gamma, \pi') - n(\gamma, s, \pi)d(\gamma, \pi)}{d(\gamma, \pi)d(\gamma, \pi')}
\end{aligned}
$$

Therefore, $v_{\gamma,s}^{\pi} - v_{\gamma,s}^{\pi'} = 0$ for $\gamma \in [0, 1)$ implies that $\gamma$ is a root of the following polynomial equation in $\gamma$:

$$p(\gamma) = 0, \tag{D.2}$$

for $p$ the polynomial defined as

$$p(\gamma) = n(\gamma, s, \pi)d(\gamma, \pi') - n(\gamma, s, \pi')d(\gamma, \pi). \tag{D.3}$$

**Step 2.** We now study the properties of the polynomial $p$. Note that it is straightforward that $p$ is a polynomial of degree $N = 2|\mathcal{S}| - 1$. We first study the properties of the polynomial $\gamma \mapsto d(\pi, \gamma)$. We have the following lemma.

**Lemma D.2.** *We have*

$$d(\gamma, \pi) > 0, \forall \gamma \in [0, 1), \forall\, \pi \in \Pi,$$

*and $d(1, \pi) = 0, \forall\, \pi \in \Pi$.*

*Proof of Lemma D.2.* This lemma follows from the relation between the determinant of a matrix and its eigenvalues, through the characteristic polynomial:

$$d(\gamma, \pi) = \det\left(\boldsymbol{I} - \gamma\boldsymbol{P}_\pi\right) = \prod_{\lambda \in Sp(\boldsymbol{P}_\pi)} (1 - \gamma\lambda)^{\alpha_\lambda},$$

with $\alpha_\lambda$ the algebraic multiplicity of the (potentially complex) eigenvalue $\lambda$ in the spectrum $Sp(\boldsymbol{P}_\pi)$ of $\boldsymbol{P}_\pi$. Since $\boldsymbol{P}_\pi$ is the transition matrix of a Markov chain, we know that the modulus of any eigenvalue $\lambda$ of $\boldsymbol{P}_\pi$ is smaller or equal to 1. This shows that $d(\gamma, \pi) > 0, \forall\, \gamma \in [0, 1), \forall\, \pi \in \Pi$. To show $d(1, \pi) = 0$, we simply note that $1 \in Sp(\boldsymbol{P}_\pi)$ since $\boldsymbol{P}_\pi$ is the transition matrix of a Markov chain. $\qquad\square$

From Lemma D.2 and the definition of $p$ as in (D.3), it is straightforward that $p(1) = 0$.

**Lemma D.3.** $\gamma = 1$ *is a root of $p$.*

We now bound the sum of the absolute values of the coefficients of $p$. We have the following theorem.

**Theorem D.4.** *The polynomial $m^{2|\mathcal{S}|} \cdot p$ has integral coefficients, potentially negative. The sum of the absolute values of the coefficients of $m^{2|\mathcal{S}|}p$ is bounded by*

$$L = 2 \cdot |\mathcal{S}| \cdot r_\infty \cdot m^{2|\mathcal{S}|} \cdot 4^{|\mathcal{S}|}.$$

Theorem D.4 is based on the following three propositions. We note $C_\ell^k$ the binomial coefficient defined as $C_\ell^k = \ell!/k!(\ell - k)!$.

**Proposition D.5.** *For any $\pi \in \Pi$, the function $\gamma \mapsto d(\pi, \gamma)$ is a polynomial of degree $|\mathcal{S}|$. Moreover, $\gamma \mapsto m^{|\mathcal{S}|} \cdot d(\pi, \gamma)$ is a polynomial with integral coefficients (potentially negative), and the absolute value of its coefficient of degree $k$ is bounded by $m^{|\mathcal{S}|}C_{|\mathcal{S}|}^k$.*

*Therefore, the sum of the absolute values of the coefficients of $\gamma \mapsto m^{|\mathcal{S}|} \cdot d(\pi, \gamma)$ is upper bounded by*

$$L_d = m^{|\mathcal{S}|} \cdot 2^{|\mathcal{S}|}.$$

**Proposition D.6.** *For any policy $\pi \in \Pi$ and any state $s \in \mathcal{S}$, the function $\gamma \mapsto n(\gamma, s, \pi)$ is a polynomial of degree $|\mathcal{S}| - 1$. Moreover, $\gamma \mapsto m^{|\mathcal{S}|} \cdot n(\gamma, s, \pi)$ is a polynomial with integral coefficients (potentially negative), and the absolute value of its coefficient of degree $k$ is bounded by $m^{|\mathcal{S}|} \cdot |\mathcal{S}| \cdot r_\infty \cdot C_{|\mathcal{S}|-1}^k \cdot 2$.*

*Therefore, the sum of the absolute values of the coefficients of $\gamma \mapsto m^{|\mathcal{S}|} \cdot n(\gamma, s, \pi)$ is upper bounded by*

$$L_n = m^{|\mathcal{S}|-1} \cdot |\mathcal{S}| \cdot r_\infty \cdot 2^{|\mathcal{S}|}.$$

**Proposition D.7.** *Let $P = \sum_{i=0}^n a_i X^i, Q = \sum_{j=0}^m b_j X^j$. Then $PQ = \sum_{k=0}^{n+m} c_k X^k$, $c_k = \sum_{i,j;i+j=k} a_i b_j$. Additionally, suppose that $\sum_{i=0}^n |a_i| \leq L_P, \sum_{j=0}^m |b_j| \leq L_Q$. Then*

$$\sum_{k=0}^{n+m} |c_k| \leq L_P L_Q.$$

Combining Proposition D.5, Proposition D.6 and Proposition D.7 with the definition of the polynomial $p$ in (D.3) yields Theorem D.4.

To conclude Step 2 of our proof, let us prove Proposition D.5 and Proposition D.6. Proposition D.7 simply follows from the multiplication rule for polynomials.

*Proof of Proposition D.5.* By definition,

$$d(\gamma, \pi) = \det\left(\boldsymbol{I} - \gamma \boldsymbol{P}_\pi\right) = \sum_{k=0}^{|\mathcal{S}|} a_k\left(\gamma \boldsymbol{P}_\pi\right),$$

where $\boldsymbol{M} \mapsto a_k\left(\boldsymbol{M}\right)$ is the $(|\mathcal{S}| - k)$-th coefficient of the characteristic polynomial of a matrix $\boldsymbol{M}$. By definition, $a_k(\boldsymbol{M})$ is the sum of all the principal minors of size $k$ of $\boldsymbol{M}$ (section 0.7.1, (Horn and Johnson, 2012)). This first shows that $a_k\left(\gamma \boldsymbol{P}_\pi\right) = \gamma^k a_k\left(\boldsymbol{P}_\pi\right)$, and therefore, that

$$d(\gamma, \pi) = \sum_{k=0}^{|\mathcal{S}|} \gamma^k a_k\left(\boldsymbol{P}_\pi\right).$$

We will show that

$$a_k(\boldsymbol{P}_\pi) \leq C_{|\mathcal{S}|}^k, \forall\, k = 1, ..., |\mathcal{S}|.$$

Let $g$ be a principal minor of $\boldsymbol{P}_\pi$ of size $k$. By definition, $g$ is the determinant of a submatrix $\boldsymbol{M}$ of size $k$ of $\boldsymbol{P}_\pi$, obtained by deleting rows and columns with the same indices: $g = \det(\boldsymbol{M})$. For any matrix square $\boldsymbol{M}$, we always have $\det(\boldsymbol{M}) = \det(\boldsymbol{M}^\top)$. Now Hadamard's inequality shows that $\det(\boldsymbol{M}^\top) \leq \prod_{i=1}^k \|Col_i(\boldsymbol{M}^\top)\|_2$, with $Col_i(\boldsymbol{M}^\top)$ the $i$-th column of $\boldsymbol{M}^\top$, and therefore we have $\det(\boldsymbol{M}^\top) \leq \prod_{i=1}^k \|Col_i(\boldsymbol{M}^\top)\|_1$. Note that the columns of $\boldsymbol{M}^\top$ have $\ell_1$-norm smaller than 1, since $\boldsymbol{P}_\pi$ is a stochastic matrix, and $\boldsymbol{M}$ is a submatrix of $\boldsymbol{P}_\pi$. Therefore, $g \leq 1$. Because there are $C_n^k$ possible principal minors of size $k$ of $\boldsymbol{P}_\pi$, we have $a_k(\boldsymbol{P}_\pi) \leq C_n^k, \forall\, k = 1, ..., n$.

Of course, we may have $a_k(\boldsymbol{P}_\pi) \notin \mathbb{Z}$. However, for any principal minor $g = \det(\boldsymbol{M})$ of $\boldsymbol{P}_\pi$, we have, by definition the determinant,

$$\det(\boldsymbol{M}) = \sum_{\sigma \in \mathfrak{S}_k} \varepsilon(\sigma) \prod_{i=1}^k M_{\sigma(i)i}$$

where $\varepsilon(\sigma)$ is the signature of the permutation $\sigma$ and $\mathfrak{S}_k$ is the symmetric group, i.e., the group of all permutations of $\{1, ..., k\}$. This shows, by definition $m$ as the maximum bit-size of the input data, that $m^{|\mathcal{S}|} \det(\boldsymbol{M}) \in \mathbb{Z}$, and therefore that $m^{|\mathcal{S}|} a_k(\boldsymbol{P}_\pi) \in \mathbb{Z}$ and that $m^{|\mathcal{S}|} a_k(\boldsymbol{P}_\pi) \leq m^{|\mathcal{S}|} C_{|\mathcal{S}|}^k$. □

*Proof of Proposition D.6.* Using Laplace cofactor expansions (section 0.3.1, (Horn and Johnson, 2012)), we have that $n(\gamma, s, \pi)$ is equal to

$$\sum_{s' \in \mathcal{S}} (-1)^{s+s'} \cdot r_{s', \pi(s')} \cdot \det\left((\boldsymbol{I} - \gamma \boldsymbol{P}_\pi)_{\mathcal{S} \setminus \{s'\} \times \mathcal{S} \setminus \{s\}}\right), \tag{D.4}$$

where $(\boldsymbol{I} - \gamma \boldsymbol{P}_\pi)_{\mathcal{S} \setminus \{s'\} \times \mathcal{S} \setminus \{s\}}$ is the matrix obtained from $\boldsymbol{I} - \gamma \boldsymbol{P}_\pi$ by removing the $s$-th column and the $s'$-th row.

Note that $\gamma \mapsto \det\left((\boldsymbol{I} - \gamma \boldsymbol{P}_\pi)_{\mathcal{S} \setminus \{s'\} \times \mathcal{S} \setminus \{s\}}\right)$ is a polynomial of degree $|\mathcal{S}| - 1$ in $\gamma$. Similarly as for the proof of Proposition D.5, $\gamma \mapsto m^{|\mathcal{S}|} n(\gamma, s, \pi)$ is a polynomial of degree $|\mathcal{S}| - 1$ with integral coefficients.

Let us consider $\boldsymbol{I}_{\setminus \{s', s\}}$ the matrix of dimension $(|\mathcal{S}| - 1) \times (|\mathcal{S}| - 1)$, obtained by removing the $s$-th column and the $s'$-th row from the identity matrix of dimension $|\mathcal{S}|$, and let us call $\boldsymbol{E}_{s'}$ the matrix of dimension $(|\mathcal{S}| - 1) \times (|\mathcal{S}| - 1)$, where all rows are $\boldsymbol{0}^\top$, except the $s$-th row, equal to $\boldsymbol{e}_{s'}^\top$.

Then $\det\left((\boldsymbol{I} - \gamma \boldsymbol{P}_\pi)_{\mathcal{S} \setminus \{s'\} \times \mathcal{S} \setminus \{s\}}\right)$ is equal to

$$\det\left((\boldsymbol{I} - \gamma \boldsymbol{P}_\pi)_{\mathcal{S} \setminus \{s'\} \times \mathcal{S} \setminus \{s\}} + \boldsymbol{E}_{s'} - \boldsymbol{E}_{s'}\right)$$

and therefore is equal to

$$\det\left(\boldsymbol{I}_{\backslash\{s',s\}} + \boldsymbol{E}_{s'} - (\gamma\boldsymbol{P}_\pi)_{\mathcal{S}\backslash\{s'\}\times\mathcal{S}\backslash\{s\}} - \boldsymbol{E}_{s'}\right).$$

We notice that $\boldsymbol{I}_{\backslash\{s',s\}} + \boldsymbol{E}_{s'}$ is a matrix whose rows are exactly the rows of the identity matrix of $\mathbb{R}^{|\mathcal{S}|-1}$, up to a certain permutation $\sigma \in \mathfrak{S}_{|\mathcal{S}|-1}$. Let $\boldsymbol{P}^\sigma \in \mathbb{R}^{(|\mathcal{S}|-1)\times(|\mathcal{S}|-1)}$ the permutation matrix defined as $P_{ij} = 1$ if $\sigma(j) = i$ and $0$ otherwise. Then for any matrix $\boldsymbol{M}$, we have $\det(\boldsymbol{P}^\sigma \boldsymbol{M}) = \det(\boldsymbol{P}^\sigma)\det(\boldsymbol{M}) = \varepsilon(\sigma)\det(\boldsymbol{M})$, with $\varepsilon(\sigma)$ the signature of the permutation $\sigma$. Since we always have $\varepsilon(\sigma) \in \{-1,1\}$, this shows that $\det\left((\boldsymbol{I} - \gamma\boldsymbol{P}_\pi)_{\mathcal{S}\backslash\{s'\}\times\mathcal{S}\backslash\{s\}}\right)$ is equal to

$$\varepsilon(\sigma)\det\left(\boldsymbol{I} - \left((\gamma\boldsymbol{P}_\pi)_{\mathcal{S}\backslash\{s'\}\times\mathcal{S}\backslash\{s\}} + \boldsymbol{E}_{s'}\right)\right).$$

The map $\gamma \mapsto \det\left(\boldsymbol{I} - \left((\gamma\boldsymbol{P}_\pi)_{\mathcal{S}\backslash\{s'\}\times\mathcal{S}\backslash\{s\}} + \boldsymbol{E}_{s'}\right)\right)$ is equal to

$$\sum_{k=0}^{|\mathcal{S}|-1} a_k\left((\gamma\boldsymbol{P}_\pi)_{\mathcal{S}\backslash\{s'\}\times\mathcal{S}\backslash\{s\}} - \boldsymbol{E}_{s'}\right)$$

where similarly as for the proof of Proposition D.5, $a_k(\boldsymbol{M})$ is the $k$-th coefficient of the characteristic polynomial of a matrix $\boldsymbol{M}$, i.e., $a_k(\boldsymbol{M})$ is equal to the sum of all the principal minors of $\boldsymbol{M}$ of dimension $k \times k$. Let

$$\boldsymbol{M} = (\gamma\boldsymbol{P}_\pi)_{\mathcal{S}\backslash\{s'\}\times\mathcal{S}\backslash\{s\}} - \boldsymbol{E}_{s'}.$$

Note that $(\boldsymbol{P}_\pi)_{\mathcal{S}\backslash\{s'\}\times\mathcal{S}\backslash\{s\}}$ is a substochastic matrix, i.e., it has non-negative entries and the sum of the entries of each row is smaller or equal to 1. Note that $\boldsymbol{M}$ differs from $(\gamma\boldsymbol{P}_\pi)_{\mathcal{S}\backslash\{s'\}\times\mathcal{S}\backslash\{s\}}$ only at the coefficient of index $(s, s')$. Using Hadamard's inequality, we find that that

$$a_k(\boldsymbol{M}) \le 2 \cdot C_{|\mathcal{S}|-1}^k, m^{|\mathcal{S}|} a_k(\boldsymbol{M}) \in \mathbb{N}. \tag{D.5}$$

We conclude by combining Equation (D.5) with Equation (D.4). $\qquad\square$

**Step 3.** We now lower bound the distance between any two roots of $p$ by a scalar $\eta > 0$. Since we know that for $\gamma(\pi, \pi', s) \in [0, 1)$ and $1$ are two roots of $P$, this will show that $\gamma(\pi, \pi', s) < 1 - \eta$.

Our proof is based on the following theorem.

**Theorem D.8** ((Rump, 1979)). *Let $p$ be a polynomial of degree $N$ with integer coefficients, possibly with multiple roots. Let $L$ be the sum of the absolute values of its coefficients. Then the distance between any two distinct roots of $p$ is strictly larger*

$$\frac{1}{2N^{N/2+2}\left(L+1\right)^N}.$$

Recall that both $\gamma(\pi, \pi', s) \in [0, 1)$ and $1$ are roots of the polynomial $p$. Therefore, we can combine Theorem D.8 with Theorem D.4 to obtain $\gamma(\pi, \pi', s) < 1 - \eta(\mathcal{M})$, with

$$\eta(\mathcal{M}) = \frac{1}{2N^{N/2+2}\left(L+1\right)^N}$$

with

$$N = 2|\mathcal{S}| - 1,$$
$$L = 2 \cdot |\mathcal{S}| \cdot r_\infty \cdot m^{2|\mathcal{S}|} \cdot 4^{|\mathcal{S}|}.$$

This concludes the proof of Theorem 4.4.

**Remark D.9.** Note that (Akian et al., 2019) use Theorem D.8 to obtain a lower bound on the average rewards of any two different policies, in the setting of two-player stochastic games.

**Remark D.10.** Theorem 1 in (Rump, 1979) provides a separation bound in the case where the polynomial $p$ has complex coefficients. Unfortunately, the separation bound from Theorem 1 in (Rump, 1979) is not directly usable here, because it depends on the *discriminant* $D(p)$ of the polynomial $p$, a quantity that is hard to lower-bound (in all generality). We decide to use the bound from Theorem 3 in (Rump, 1979) because it does not depend on $D(p)$ but directly on the $\ell_1$-norm of $p$ and of the degree of $p$, which can be computed in closed-form and can be bounded as in Proposition D.6 and Proposition D.5.

# E    Proof of Theorem 4.7

*Proof of Theorem 4.7.* Following table 4 in (Ye, 2011), we know that interior-point methods for the linear programming formulation of MDPs return an optimal policy in $O\left(|\mathcal{S}|^3|\mathcal{A}|^2\left(Q(\boldsymbol{r}, \boldsymbol{P}, \gamma)\right)\right)$ arithmetic operations, with $Q(\boldsymbol{r}, \boldsymbol{P}, \gamma)$ equal to the total bit-size of the MDP instance, i.e., the sum of the bit-sizes of all instantaneous rewards, transition probabilities, and the discount factor. By choosing $\gamma = 1 - \eta(\mathcal{M})$ and noticing that $\log(\eta(\mathcal{M})) = O\left(|\mathcal{S}|\log(r_\infty) + |\mathcal{S}|^2\log(m)\right) = O\left(|\mathcal{S}|^2\log(m)\right)$, we see that interior-point methods for the linear programming formulation of MDPs return an optimal policy in $O\left(|\mathcal{S}|^5|\mathcal{A}|^2\left(Q(\boldsymbol{r}, \boldsymbol{P})\right)\right)$, where $Q(\boldsymbol{r}, \boldsymbol{P})$ is the total bit-size of MDP instance. $\qquad\square$

# F    Proof of Section 4.2

*Proof of the existence of $\gamma_{\mathsf{bw,r}}$.* Let

$$\bar{\gamma}_{\mathsf{r}} = \max_{\pi, \pi' \in \Pi, s \in \mathcal{S}} \max_{\boldsymbol{P}, \boldsymbol{P}' \in \mathcal{U}_{\mathrm{ext}}} \gamma(\pi, \pi', s, \boldsymbol{P}, \boldsymbol{P}'),$$

where $\gamma(\pi, \pi', s, \boldsymbol{P}, \boldsymbol{P}')$ is the largest zero of the function $\gamma \mapsto v_{\gamma,s}^{\pi,\boldsymbol{P}} - v_{\gamma,s}^{\pi',\boldsymbol{P}'}$ if it is not identically equal to zero, or $\gamma(\pi, \pi', s, \boldsymbol{P}, \boldsymbol{P}') = 0$ otherwise. Recall that $\mathcal{U}_{\mathrm{ext}}$ is the (finite) set of extreme points of $\mathcal{U}$. We will show that $\Pi_{\gamma,\mathsf{r}}^\star = \Pi_{\mathsf{bw,r}}^\star, \forall\, \gamma > \bar{\gamma}_{\mathsf{r}}$. Let $\pi$ be a robust discount-optimal policy for some $\gamma > \bar{\gamma}_{\mathsf{r}}$. We will prove that $\pi$ is a Blackwell-optimal policy. Since $\pi$ is robust $\gamma$-discount-optimal, we have

$$v_{\gamma,s}^{\pi,\mathcal{U}} \geq v_{\gamma,s}^{\pi',\mathcal{U}}, \forall\, \pi' \in \Pi, \forall\, s \in \mathcal{S}.$$

By definition $v_{\gamma,s}^{\pi,\mathcal{U}} = \min_{\boldsymbol{P} \in \mathcal{U}} v_{\gamma,s}^{\pi,\boldsymbol{P}}, \forall\, s \in \mathcal{S}$. From (Iyengar, 2005), we know that the $\arg\min$ in $\min_{\boldsymbol{P} \in \mathcal{U}} v_{\gamma,s}^{\pi,\boldsymbol{P}}$ is attained at an extreme point of $\mathcal{U}$. Therefore, by definition of $\bar{\gamma}_{\mathsf{r}}$, the function $\gamma \mapsto v_{\gamma,s}^{\pi,\mathcal{U}} - v_{\gamma,s}^{\pi',\mathcal{U}}$ cannot be equal to 0 on $(\bar{\gamma}_{\mathsf{r}}, 1)$, and therefore it does not change sign, since it is a continuous function. This shows that for all $\gamma > \bar{\gamma}_{\mathsf{r}}$, we have

$$v_{\gamma,s}^{\pi,\mathcal{U}} \geq v_{\gamma,s}^{\pi',\mathcal{U}}, \forall\, \pi' \in \Pi, \forall\, s \in \mathcal{S}.$$

This shows the existence of the robust Blackwell discount factor $\gamma_{\mathsf{bw,r}}$ and that $\gamma_{\mathsf{bw,r}} < \bar{\gamma}_{\mathsf{r}}$. $\qquad\square$

*Proof of Theorem 4.11.* We start by showing the following lemma.

**Lemma F.1.** *Let $\mathcal{M} = \left(\mathcal{S}, \mathcal{A}, \boldsymbol{r}, \boldsymbol{P}^0\right)$ be an MDP instance with maximum bit-size $m \in \mathbb{N}$. Assume that $\mathcal{U}$ is sa-rectangular, where for each $(s, a) \in \mathcal{S} \times \mathcal{A}$, $\mathcal{U}_{sa}$ is constructed as in* (4.1)*, with the scalars $(\alpha_{sa})_{s,a}$ of maximum bit-size $m$.*

*Then the maximum bit-size complexity to describe the transition probabilities associated with the extreme points of $\mathcal{U}_{sa}$ is $m'$ for $p = \infty$ and $2m'$ for $p = 1$.*

*Proof of Lemma F.1.* In the proof of this lemma, we use the fact that the worst-case kernel $\boldsymbol{P}^\star$ of a policy $\pi$ can be chosen as the $\arg\min$ of the optimization problem $\min_{\boldsymbol{p} \in \mathcal{U}_{s\pi(s)}} \boldsymbol{p}^\top \boldsymbol{v}_\gamma^{\pi,\mathcal{U}}$, where $\boldsymbol{v}_\gamma^{\pi,\mathcal{U}}$ is the worst-case value function of $\pi$. In particular, let $\boldsymbol{v} \in \mathbb{R}^{\mathcal{S}}$.

**The case $p = \infty$.** In this case, there exists a sorting solution to $\min_{\boldsymbol{p} \in \mathcal{U}_{sa}} \boldsymbol{p}^\top \boldsymbol{v}$ for any $(s, a) \in \mathcal{S} \times \mathcal{A}$ and any $\boldsymbol{v} \in \mathbb{R}^{\mathcal{S}}$, by sorting $\boldsymbol{v}$, see for instance proposition 3 in (Goh et al., 2018), equation (9) in (Givan et al., 1997), or appendix C in (Behzadian et al., 2021). In particular, let $(s, a) \in \mathcal{S} \times \mathcal{A}$ and define $\sigma$ the permutation of $\mathcal{S}$ such that $v_{\sigma(1)} \leq \ldots \leq v_{\sigma(|\mathcal{S}|)}$, and define $i$ as the smaller integer in $\{1, \ldots, |\mathcal{S}|\}$ such that

$$\sum_{s'=1}^{i}\left(P_{sa\sigma(s')}^0 + \alpha_{sa}\right) + \sum_{s'=i+1}^{|\mathcal{S}|}\left(P_{sa\sigma(s')}^0 - \alpha_{sa}\right) \geq 1.$$

Then a solution to $\min_{\boldsymbol{p} \in \mathcal{U}_{sa}} \boldsymbol{p}^\top \boldsymbol{v}$ is $p_{\sigma(s')} = P^0_{sa\sigma(s')} + \alpha_{sa}$ if $s' < i$, $p_{\sigma(s')} = P^0_{sa\sigma(s')} - \alpha_{sa}$ if $s' > i$, and

$$p_{\sigma(i)} = 1 - \sum_{s' \in \mathcal{S} \setminus \{i\}} p_{\sigma(s')}.$$

This closed-form shows that for any vector $\boldsymbol{v} \in \mathbb{R}^{\mathcal{S}}$, a solution of $\min_{\boldsymbol{p} \in \mathcal{U}_{sa}} \boldsymbol{p}^\top \boldsymbol{v}$ can be found as a vector with rational entries with a denominator of at most $m$.

**The case $p = 1$.** In this case, one can show that the optimization problem $\min_{\boldsymbol{p} \in \mathcal{U}_{sa}} \boldsymbol{p}^\top \boldsymbol{v}$ can be formulated as a linear program. Therefore, there exists an optimal basic feasible solution $\boldsymbol{p}$ which has the following form by lemma 5.4 and lemma 5.5 in (Ho et al., 2021). There exist $j_1, j_2 \in \mathcal{S}$ such that $j_1 \neq j_2$ and for each $i \in \mathcal{I} = \mathcal{S} \setminus \{j_1, j_2\}$:

$$p_i = 0 \quad \text{or} \quad p_i = P^0_{sai}$$
$$p_{j_1} \geq P^0_{saj_1} \quad \text{and} \quad p_{j_2} \leq P^0_{saj_2}.$$

Then, in order for $\boldsymbol{p} \in \mathcal{U}_{sa}$ we need the following equalities to hold

$$p_{j_1} + p_{j_2} = 1 - \sum_{i \in \mathcal{I}} p_i$$
$$(p_{j_1} - P^0_{saj_1}) + (P^0_{saj_2} - p_{j_2}) = \alpha_{sa} - \sum_{i \in \mathcal{I}} |p_i - P^0_{sai}|.$$

Combining the equalities above yields that

$$2p_{j_1} = \alpha_{sa} - \sum_{i \in \mathcal{I}} |p_i - P^0_{sai}| + P^0_{saj_1} - P^0_{saj_2}$$
$$+ 1 - \sum_{i \in \mathcal{I}} p_i.$$

Because the right-hand side of the equation above is a sum of rational numbers with a denominator of at most $m$, $p_{j_1}$ is also rational with a denominator at most $2m$. Using an analogous argument for $p_{j_2}$, we get that there exists an optimal solution that is rational with a denominator of at most $2m$. $\quad\square$

Theorem 4.11 then follows by applying Theorem 4.4 with on the MDP instance $(\mathcal{S}, \mathcal{A}, \boldsymbol{r}, \boldsymbol{P}')$ with $\boldsymbol{P}'$ an extreme point of $\mathcal{U}$. Lemma F.1 exactly describes the maximum bit-size of any transition $P'_{sas'}$ for $(s, a, s') \in \mathcal{S} \times \mathcal{A} \times \mathcal{S}$ in the case of sa-rectangular uncertainty set based on $\ell_1$-distance or $\ell_\infty$-distance as in (4.1). This concludes the proof of Theorem 4.11. $\quad\square$

