# OpenReview forum: "Reducing Blackwell and Average Optimality to Discounted MDPs via the Blackwell Discount Factor"
_NeurIPS.cc/2023/Conference — NeurIPS 2023 poster_

### Official Review · Reviewer_TMw8 · 2023-06-16

**Soundness:** 3 good
**Presentation:** 2 fair
**Contribution:** 2 fair
**Rating:** 5
**Confidence:** 2

**Summary:**

The paper introduces the Blackwell discount factor MDPs.
The authors show that when the discount factor is larger than the Blackwell discount factor $\gamma_{bw}$, all discount-optimal policies become Blackwell- and average-optimal, and derive a general upper bound on $\gamma_{bw}$. The authors claim to provide the first reduction from average and Blackwell optimality to discounted optimality via polynomial-time algorithms.

I am not much familiar with the literature regarding Blackwell optimality criterion and thus might not fully understand the significant of the results. I would be happy that the authors would highlight for significant of results in compression to the known results in for discounted value criterion and averaged value criterion.



**Strengths:**


1)	The assumptions and contributions are clearly stated.
2)	It seems that the presented results improve the known results in the study of Blackwell optimality criterion, remove commonly used assumptions and provide efficient algorithms.
3)	The application of algebraic techniques to RL might be useful in other settings.


**Weaknesses:**


1)	No concrete application or motivation for the work is given.
2)	It is unclear to me whether the new definition of Blackwell optimality via the Blackwell discount factor proposed in the paper is better/ outperform/ more useable than the discounted value criterion in standard RL applications.
3)	The authors claim throughout the paper that previous literature repeatedly mentioned the discussed problem as important, but not clearly mentioned why.
4)	The focus of the paper is mainly about previous works (and their limitations),  instead of the new approach (they get to it only in page 6 out of 9!). More elaborated proof sketches would benefit the reader.


**Questions:**

1)	May you please provide concrete motivation for the use in Blackwell optimality criterion and/or a concrete application?
2)	Ergodicity and aperiodicity properties are standard assumption in RL even in simple planning task. It is unclear to me how can you remove them and why it is important. In most application the assumptions are reasonable.

---

> ### Author Rebuttal · Authors · 2023-08-09
>
> We thank you for your time reviewing the paper and for your constructive comments. We answer your questions and remarks below.
>
> *Motivation and applications of Blackwell optimality.* The motivation for Blackwell optimality mostly comes from some of the shortcomings of the two other return criteria.
>
> * The discounted return criterion enjoys nice tractability properties: it can be solved efficiently *without any structural assumptions on the MDP*. However, this requires choosing a value for the discount factor $\gamma$, which may be artificial in some applications. Additionally, the optimal policies may change with the discount factors. These issues are well-documented in the RL/MDP community [1].
>
> * For this reason, *average optimality* is getting some attention in the RL literature [2], especially since many RL applications require extremely long horizons, e.g. in game-solving (the game may end after thousands of decisions) or in healthcare (the health condition of the patient in a far horizon remains an important concern). However, average optimality suffers from its own shortcomings: (a) the vast majority of the literature requires some structural assumptions (unichain, ergodicity, weak-communication, etc.), mostly motivated from a technical standpoint; (b) the case of adversarial/robust MDP is still not very well understood for average optimality; (c) most importantly, average optimality discards {\em any} rewards obtained in a finite horizon. So it does not distinguish between a policy $\pi_{1}$ achieving rewards $0 + 0 + 0 + 0 + ... + 1 + 1 + ...$, where the $0$'s are repeated a $1000$ times before switching to $1$ forever, and a policy $\pi_{2}$ achieving rewards $1 + 1 + ...$ which always obtains $1$. Clearly, $\pi_{2}$ is a better policy, but it attains the same average reward as $\pi_{1}$.
>
> * This very shortcoming of average optimality is addressed by Blackwell optimality. Indeed, since a Blackwell optimal policy is optimal for all discount factors close to $1$, it also takes into account the rewards obtained in finite times. Additionally, there is no need to choose a specific value of the discount factor, in contrast to discount optimality. In fact, one can show that Blackwell optimal policies are average optimal but also optimizes the {\em bias} (i.e., the rewards obtained in the transient regime where the Markov chain converges to its stationary distribution), and higher-order terms, as explained in Section 10.2 in [3]. Still, although six decades have passed since Blackwell optimality was introduced in [4], there are still no efficient algorithms to compute a Blackwell optimal policy. As detailed in our paper, all existing algorithms are so complicated that we are not aware of any practical implementations of these ideas. We view our work as a first step toward a tractable approach to Blackwell optimality by providing an algorithm (solving discount optimal policy for $\gamma > \gamma_{\sf bw}$) that is conceptually much simpler than previous methods. Our Theorem 4.7 shows that our algorithm is weakly-polynomial, matching the most widely accepted class of {\em efficient algorithms. We would also like to emphasize that we are the first to show the shortcomings of the existing definition and to clarify some misunderstandings (see page 5, Proposition 3.4).}
>
>  We will use the additional page available for the final version to clarify this and highlight the positioning of our work.
>
>  [1] Y. Tang, M. Rowland, R. Munos, and M. Valko. Taylor expansion of discount factors. ICML 2021.
>
>  [2] V. Dewanto, G. Dunn, A. Eshragh, M. Gallagher, and F. Roosta. Average-reward model-free reinforcement learning: a systematic review and literature mapping.
>
>  [3] M. L. Puterman. Markov Decision Processes: Discrete Stochastic Dynamic Programming. John Wiley and Sons, 2014.
>
>  [4] D. Blackwell. Discrete dynamic programming. The Annals of Mathematical Statistics, pages 719–726, 1962.
>
> *Common structural assumptions such as ergodicity/weak-communication*. Our goal is to study Blackwell optimality (and consequently average optimality) in all generality. In the course of this rebuttal, we have noticed that our Example 3.5 (shortcomings of the existing definitions of Blackwell optimality) and Proposition 4.3 (need for ``coarseness'' conditions to bound $\gamma_{\sf bw}$) can be extended to the case of weakly-communicating and unichain MDPs. Therefore, the issues identified in this paper still remain under some of the most common structural assumptions on the MDP instance. We refer to the PDF attached to our response common to all reviewers for the new MDP instances and to our response to Reviewer mmfD for the detailed computations.
>
> ### Conclusion.
> We thank you for your time reviewing the paper and your detailed questions, which will lead to an improved manuscript. We are looking forward to hearing back from you and we hope that our responses convince you of the completeness and significance of our paper and lead to an increased score.

---

> > ### Comment · Reviewer_TMw8 · 2023-08-10
> >
> > I thank the authors for their response and have no further questions.

---

### Official Review · Reviewer_ePug · 2023-07-06

**Soundness:** 3 good
**Presentation:** 3 good
**Contribution:** 3 good
**Rating:** 5
**Confidence:** 3

**Summary:**

This paper studied the a Blackwell optimal policy can be obtained from an average-optimal policy by introducing the Blackwell discount factor $\gamma_{bw}$. The author further showed that the discount factor can be upper bounded by the bit-size of an MDP instance.

**Strengths:**

1. The paper is well written and organized.  The insights behind the main results are well presented.
2. The introducing of the Blackwell discount factor is interesting, it may lead to efficient algorithms in future studies.
3. An upper bound for $\gamma_{bw}$ is provided.

**Weaknesses:**

1. Providing the upper bound of $\gamma_{bw}$ is not enough, still we don’t have a efficient algorithm for finding the Blackwell policy. Also I am afraid for calculating it needs have sufficient information of the MDP structure.
2. Proposition 4.3 is only for the deterministic transitions with $S=2, A=2.$
3. The $ \eta$ in Theorem is quite close to zero, which means the upper bound for $\gamma_{bw}$ is almost 1.
4. No simulation results are provided, it is better the show the advantage of the proposed approach with even some simple simulations.

**Questions:**

1. I don’t agree with the authors that they criticize on the existing definition of the Blackwell optimality. The definition looks good to me, we just don’t have an efficient algorithm yet. What if the MDP is unichain or ergodic (different as your example 3.5), does proposition 3.4 still hold?  As far as I know, the weakly communicating assumption or the ergodic assumption, is in fact known to be necessary for learning infinite-horizon MDPs with low regret.
2. what is m in your theorem 4.4?
3. Proposition 4.3: \eta >0 → 1>\eta>0

**Limitations:**

More discussions on the limitations of your approach should be included. Also it is better to make more comparisons between the two discounted factors when there are more assumptions on the MDP.

---

> ### Author Rebuttal · Authors · 2023-08-09
>
>
> Thank you for your review. We first answer the weaknesses.
>
> 1. Note that our bound on $\gamma_{\sf bw}$ leads to a *polynomial* algorithm; we refer to Theorem 4.7 for the complexity statement. Polynomial-time complexity is commonly understood as {\em tractable} empirical performances. Our bound requires the knowledge of the number of states and actions, and the *maximum bit-size* of the MDP. This bit size is simply determined by (a) an upper bound of the rewards and (b) by the distance of positive transition probabilities to 0. The term (a) can be estimated since we only need an upper bound. The term (b) can be estimated when we build the transition probabilities as the empirical frequencies of the observed transitions from historical data, i.e., as $P_{sas'} = n_{sas'}/n_{sa}$ with $n_{sas'}$ the number of observation of the transition $s,a$ to the next state $s'$ and $n_{sa}$ the number of observation the state-action pair $s,a$.
>
> 2. We wrote Proposition 4.3 with the simplest MDP instance. We can add fictitious states/actions and non-deterministic transitions to this instance to show that the same conclusion holds with more states/actions/complicated transitions. We can also extend Proposition 4.3 to weakly-communicating MDPs. For this, we add a deterministic transition from $s_2$ to $s_1$, with a reward of $0$ for $a_{1}$ and a reward of $\epsilon$ for $a_{2}$ (see Figure 2 in the PDF attached to our response common to all reviewers). This MDP is weakly-communicating: $\{s_{1},s_{2}\}$ is strongly connected for $a_{2}$. We have $v_{\gamma}^{a_{1}} = 0, v_{\gamma}^{a_{2}} = (-1+\epsilon \gamma)/(1-\gamma)$. Thus $a_2$ is Blackwell optimal for $\gamma \geq 1/\epsilon$. With $\epsilon \geq 1, \epsilon \rightarrow 1$, we have $\gamma_{\sf bw} \rightarrow 1$. This extends Proposition 4.3 to unichain MDPs. We thank you for raising this interesting point, we will add this new result.
>
> 3. *The $\eta$ in Theorem is close to zero.*
>
> Assuming that you are referring to Theorem 4.4, we agree that the bound on $\gamma_{bw}$ is close to 1, which complicates the practical use of this result. However, we would like to clarify that $1-\gamma_{bw}$ is sufficiently large to have a polynomial sized representation. This fact is used to construct a polynomial-time algorithm in Theorem 4.7.
>
> 4. *Experimental validation.*
>
> We see our primary contribution as a better understanding of Blackwell optimality in nominal/robust MDPs. We are the first to highlight the notion of the Blackwell discount factors as a more refined tool to obtain Blackwell optimal and average optimal policies, whereas many papers before us were incorrect about when Blackwell optimal policies are optimal or not (see the discussion lines 282-296). We are also the first to show that we can use tools from polynomial analysis to study Blackwell optimality. Crucially, our paper is the first to provide a very simple algorithm to compute Blackwell/average optimal policies (through discounted MDPs), even though this notion has been around since the 1960s (see the last paragraph before section 3.2). We leave obtaining a more refined upper bound as future work. We conclude by noting that most of the recent works in this area do not focus directly on the implementation of the proposed algorithms [1,2,3,4,5]. Our work is a first step toward advancing the methodology and the understanding of an understudied optimality criterion (Blackwell optimality).
>
> [1] Wang, Wang, and Yang. Near Sample-Optimal Reduction-based Policy Learning for Average Reward MDP.
>
> [2] Jin and Sidford. Efficiently solving MDPs with stochastic mirror descent. ICML 2020.
>
> [4] Wang Primal-Dual $\pi $ Learning: Sample Complexity and Sublinear Run Time for Ergodic Markov Decision Problems. 2017
>
> [5] Chen and Wang. Stochastic primal-dual methods and sample complexity of reinforcement learning. 2016.
>
> ### Questions.
>
> *Q1.* While the existing definition has been introduced for more than 60 years, there exists no public implementation of any algorithm to compute a Blackwell optimal policy; the reasons for this being that algorithms for this task are all quite involved and impractical, see lines 209. Therefore, while the definition of Blackwell optimality is important, it does not lead to efficient algorithms. This is why we see our introduction of the Blackwell discount factor as an important step toward a tractable approach to Blackwell optimality, which has been missing for more than six decades.
>
> *Extending Proposition 3.4 to unichain/ergodic MDPs:* We can extend Example 3.5 to a unichain MDP as follows: we add a transition from state $7$ to state $0$, with a reward of $0$. We also add three intermediate states from $0$ to $7$ for action $a_{1}$, so that it takes as many periods to reach state $7$ from state $0$ for the three action s$a_{1},a_{2},a_{3}$. This new MDP is unichain; see Figure 1 in the PDF attached to our response common to all reviewers.
> We have $v_{\gamma}^{a_{1}} = 1/(1-\gamma^5), v_{\gamma}^{a_{2}} = (r_{1}\gamma + r_{2}\gamma^2)/(1-\gamma^5), v_{\gamma}^{a_{3}} = (r_{4}\gamma + r_{5}\gamma^2)/(1-\gamma^5)$, which are the same expressions as for Example 3.5, up to the denominator $(1-\gamma^5)^{-1}$. Therefore, we proved Proposition 3.4 for unichain MDPs. We thank you for raising this interesting point, we will mention this new result in our final version.
>
> 2. The integer $m$ is the maximum bit-size of the MDP instance, defined line 319. Intuitively it is the $\log_2$ of the denominator in the fractional representation of the rewards $r_{sa}$ and the transitions $P_{sas'}$. More details on this are given in Appendix B. For instance, for Riverswim we have $m=14$ since the maximum reward is $10,000$.
>
> 3. We will clarify that $\eta \in (0,1)$, thanks.
>
> ### Conclusion.
> We thank you for your detailed questions, which lead to an improved manuscript. We hope that our responses convince you of the completeness and significance of our paper and lead to an increased score.

---

> > ### Comment · Reviewer_ePug · 2023-08-16
> >
> > I thank the authors for their detailed response. I would like to increase the score.

---

> > > ### Author Response · Authors · 2023-08-16
> > >
> > > We thank you for reading our response and increasing your score.

---

### Official Review · Reviewer_LSPb · 2023-07-10

**Soundness:** 3 good
**Presentation:** 4 excellent
**Contribution:** 3 good
**Rating:** 7
**Confidence:** 3

**Summary:**

The paper studies how to reduce the computation of Blackwell-optimal policies to that of discount-optimal policies. The authors introduce the notion of "Blackwell discount factor", a value $\gamma_{bw} \in [0,1)$ s.t. any discount-optimal policy for $\gamma > \gamma_{bw}$ is also Blackwell optimal. They show that $\gamma_{bw}$ always exists for finite MDPs and upper bound it as a function of the size of the MDP. Finally, they extend these results to robust MDPs.

**Strengths:**

1. The paper studies a relevant problem, that of computing Blackwell optimal policies, which hasn't received much research attention so far (eg, as compared to other popular optimality criteria)
2. The contribution is significant and, to my knowledge, novel. Being able to reduce the computation of Blackwell optimal policies to discounted MDPs is absolutely relevant given the theoretical understanding and efficient algorithms that we currently have for the latter setting
3. The paper is extremely well written. Despite the complexity of the studied topic (which is also not very well known in the RL community), the authors did a very nice job in providing the necessary basics and intuitively discussing all formal results, while providing comprehensive proof sketches. Related works are also discussed with sufficient detail.

**Weaknesses:**

1. While it is nice to have an upper bound on $\gamma_{bw}$ that depends only on quantities known by the learner, the quantity $\eta(M)$ is exponentially small in the size of the MDP (eg in the number of states). This essentially means that, while we can solve a discounted MDP with $\gamma = 1-\eta(M)$ to compute a Blackwell optimal policy, we need to use a discount factor which is extremely close to 1 even for MDPs of moderate size. In practice, such a large value of $\gamma$ may simply make the learning process extremely inefficient. It is thus natural to wonder how conservative the proposed bound is. For instance, do the authors believe that there exist MDPs with a "simple structure" in which $\gamma_{bw}$ is way smaller than the stated bound?
2. There is no numerical simulation. It would have been nice to see some experiments comparing (eg in terms of run-time) existing algorithms for computing Blackwell optimal policies to a method for solving discount MDPs with the (upper bound on the) Blackwell discount factor

**Questions:**

See above.

**Limitations:**

Limitations have been discussed. No potential negative societal impact

---

> ### Author Rebuttal · Authors · 2023-08-09
>
>
> We thank you for your detailed comments. We will answer your questions below.
>
> *Q1 - conservativeness of our bound on $\gamma_{\sf bw}$.* You are correct that our bound on $\gamma_{\sf bw}$ may be loose. In particular, it relies on Theorem 4.6 (separation between the different roots of a polynomial), which may not be tight for the specific polynomials related to the value functions of the MDP instance. Even though the current approach results in a weakly-polynomial time algorithm, improving this bound is certainly an important future direction. We also note that Proposition 4.3 shows that $\gamma_{\sf bw}$ may be very close to $1$ even in some extremely simple MDP instances (two actions, one absorbing state, one non-absorbing state, deterministic transitions). Also, we can extend Proposition 3.4 (which shows the shortcoming of the existing definition of Blackwell optimality) to the case of unichain MDPs. This can be done as follows.
>
> *Extending Proposition 3.4/Example 3.5 to unichain/ergodic MDPs:* We can extend Example 3.5 to a unichain MDP as follows: we add a transition from state $7$ to state $0$, with a reward of $0$. We also add three intermediate states from $0$ to $7$ for action $a_{1}$, so that it takes as many periods to reach state $7$ from state $0$ for the three action s$a_{1},a_{2},a_{3}$. Note that this new MDP is unichain. We refer to Figure 1 in the PDF attached to our response common to all reviewers.
>
> Additionally, for this new MDP instance, we have $v_{\gamma}^{a_{1}} = 1/(1-\gamma^5), v_{\gamma}^{a_{2}} = (r_{1}\gamma + r_{2}\gamma^2)/(1-\gamma^5), v_{\gamma}^{a_{3}} = (r_{4}\gamma + r_{5}\gamma^2)/(1-\gamma^5)$, which are exactly the same expression as for Example 3.5, up to the common denominator $(1-\gamma^5)^{-1}$. Therefore, we have proved that the conclusion of Proposition 3.4 holds for unichain MDPs. We thank you for raising this interesting point; we will use the additional page available for our final version to mention this new result.
>
> We are also able to extend Proposition 4.3 to weakly-communicating MDPs. We refer to our response to reviewer mmfD for the exact computations and to Figure 2 in the PDF attached to our response common to all reviewers. We will add these new results in our revised manuscript.
>
> *Q2. Experimental validation.*
>
> We see our primary contribution as a better theoretical understanding of Blackwell optimality in (nominal MDPs and robust MDPs. We are the first to highlight the notion of the Blackwell discount factors as a more refined tool to obtain Blackwell optimal and average optimal policies, whereas many papers before us were incorrect about when Blackwell optimal policies are optimal or not (see the discussion lines 282-296). We are also the first to show that we can use tools from polynomial analysis to study Blackwell optimality. Finally, our paper is the first to provide a very simple algorithm to compute Blackwell/average optimal policies (through discounted MDPs), even though the notion of Blackwell optimality has been around since the 1960s (see the last paragraph before section 3.2). We leave obtaining a more refined and implementable upper bound as future work.
>
> We conclude this remark by noting that most of the recent and notable works in this area do not focus directly on the implementation of the proposed algorithms, e.g. [1,2,3,4,5]. We see our work as a first step toward advancing the methodology and the understanding of an understudied optimality criterion (Blackwell optimality).
>
> [1] Wang, Jinghan, Wang, Mengdi, et Yang, Lin F. Near Sample-Optimal Reduction-based Policy Learning for Average Reward MDP. arXiv preprint arXiv:2212.00603, 2022.
>
> [2] Jin, Yujia et Sidford, Aaron. Efficiently solving MDPs with stochastic mirror descent. ICML 2020.
>
> [4] Wang, Mengdi. Primal-Dual $\pi $ Learning: Sample Complexity and Sublinear Run Time for Ergodic Markov Decision Problems.arXiv preprint arXiv:1710.06100, 2017.
>
> [5] Chen, Yichen et Wang, Mengdi. Stochastic primal-dual methods and sample complexity of reinforcement learning. arXiv preprint arXiv:1612.02516, 2016.
>
> ### Conclusion.
> We thank you for your detailed questions. We are looking forward to hearing your new thoughts, and we hope that we have answered your questions and concerns and that you will consider increasing your score.

---

> > ### Comment · Reviewer_LSPb · 2023-08-11
> > **Acknowledgement**
> >
> > I thank the authors for their response. After going through all reviews and rebuttals, I still believe that the paper provides a significant contribution, though with some limitations. I will thus keep my initial view.

---

### Official Review · Reviewer_MXst · 2023-07-19

**Soundness:** 3 good
**Presentation:** 3 good
**Contribution:** 3 good
**Rating:** 7
**Confidence:** 4

**Summary:**

The paper proposed a new concept called Blackwell discount factor $\gamma_{bw}$, which enjoys good properties: any policy that is $\gamma_{bw}$-discount-optimal will be Blackwell optimal as well. If $\gamma_{bw}$ is known, then one can reduce the problem of find average-optimal policy to finding discount-optimal policies, which is practically easier. The authors provide theoretical justifications for Blackwell discount factor and a proper instance-dependent upper bound. In the end, they extend the results to Robust MDP setting, where the definitions and results continue to hold.

**Strengths:**

The paper tackles a significant shortcoming in the previous literature on Blackwell optimality: the current definition does not allow us to find Blackwell optimal policy in a naive way that is to choose a large enough $\gamma$ and simply find its discount optimal policy. The paper has a clear and strong motivation to study the problem with a minor question I will mention in the Weakness section.

The theoretical results are clearly delivered and significant. Especially some impossible results in Proposition 3.4 and Theorem 3.6 are intriguing and it reveals something fundamental about the Blackwell optimality.


**Weaknesses:**

Certain writings and motivation may be taken care:
1. Line 5: computing average-optimal policies requires only the weakly-communication assumption (instead of unichain or ergodicity), which is a reasonable assumption as weakly-communication is necessary for a unique optimal average rewards.
2. Line 43-61: There exists many value iteration algorithm for average-reward setting. I don't fully understand the motivation to find the optimal average-optimal policy through finding discounted-optimal policies. The well-known [UCRL2](https://proceedings.neurips.cc/paper_files/paper/2008/file/e4a6222cdb5b34375400904f03d8e6a5-Paper.pdf) calls value iteration as a sub-routine.
3. Saying the classical definition of Blackwell optimality has shortcomings can be confusing, because the paper does not have a new definition for Blackwell optimality. Instead, it proposes a new concept named Blackwell discount factor.
4. I am not fully convinced that Blackwell discount factor is the only discount factor of interest. In some circumstances, we may want to find the Blackwell optimal policy that is $\gamma$-discount-optimal for as small as possible $\gamma$.
5. I don't see a strong connection between Blackwell optimality and Robust MDP. It is nice to see that similar results continue to hold for Robust MDP. However, the two topics seem orthogonal to me. Before reading Section 4.2, I was expecting to see results on how Blackwell optimality improves robustness, since Blackwell optimality in general sense is one type of robustness w.r.t. the change of discount factors.

Minor comments:
1. Line 247: the authors mentioned “in the next proposition”, while it is a Theorem that follows.
2. Theorem 4.5, Theorem 4.6 should be stated as a Lemma.

**Questions:**

1. Is $\gamma_{bw} = \max_{\pi} \gamma(\pi)$? Intuitively, this makes sense as if $\gamma_{bw} < \gamma(\pi')$, then $\pi'$ is not $\gamma_{bw}$-discount-optimal, which violates the definition. If this is true, then can I understand the paper as finding the smallest $\gamma$ such that all Blackwell optimal policies are simultaneously optimal on this $\gamma$?
2. Is it NP-Hard to verify weakly-communicating assumption?

**Limitations:**

The authors mentioned approximate Blackwell optimality and robust Blackwell discount factor for other types of uncertainty sets. No societal impacts that need to be addressed.

---

> ### Author Rebuttal · Authors · 2023-08-09
>
> We thank you for your time reviewing the paper and for your constructive comments. We will answer your questions and remarks below.
>
> ### Writing and motivations.
>
> *Points 1-2-3*. We will make these two points more explicit in our final version. Thanks for the clarification.
>
> *Point 4*. Thanks for raising this interesting point. One key property of Blackwell optimal policies (that we did not highlight in the paper) is that *their value functions coincide for all $\gamma \in (0,1)$ for all states $s$*. This is because their value functions must coincide on an entire interval close enough to $1$, for instance, for $\gamma \in (\gamma_{\sf bw},1)$. Since the value functions are rational functions (ratios of polynomials), if they are equal for an infinite number of discount factors, then they are equal on the entire interval $(0,1)$. Therefore, all Blackwell optimal policies are $\gamma$-discount optimal (or not) for the same set of discount factors. We will add this discussion in the revised version of our manuscript.
>
> *Point 5. Connection with robust MDPs.* We have worked to extend our results on Blackwell optimality to the case of robust MDPs for mostly two reasons. First, from a practical standpoint, this allows us to (partially) address the situation where the data of the MDP instance (rewards and/or transition probabilities) is only partially known. Second, from a methodological standpoint, there has been some recent interest in studying average optimality for the case of robust MDPs [1,2]. Since Blackwell optimality bridges the gap between discounted optimality and average optimality for *nominal* MDPs, we believe that a better understanding of Blackwell optimality for *robust* MDPs is a first step toward new insights for the case of average optimality in robust MDPs as well.
>
> [1] Wang, Yue et al. Robust average-reward Markov decision processes. AAAI 2023.
>
> [2] Wang, Yue et al. Model-Free Robust Average-Reward Reinforcement Learning. ICML 2023.
>
> ### Questions.
>
> 1. It is *not* the case that $\gamma_{\sf bw} = \max_{\pi} \gamma(\pi)$. This is discussed in the paragraph line 282 - 296 (specifically, line 292-294): in Example 3.5, we have $\gamma_{\sf bw} = 3/4$, but there is a single Blackwell optimal policy $a_{1}$ and $\gamma(a_{1}) = 1/2$. This discrepancy is what makes introducing the notion of the Blackwell discount factor one of our important contributions.
>
> 2. Verifying that a nominal MDP is weakly communicating can be done in polynomial time; see Algorithm 4 and Theorem 3.5 in [3]. The unichain property is NP-hard to verify for nominal MDPs; see [4]. We are not aware of any algorithm for verifying the weakly-communicating property for robust MDPs. We will make this clearer in our manuscript.
>
> [3] Kallenberg, L. C. M. Classification problems in MDPs. Markov processes and controlled Markov chains, 2002, p. 151-165.
>
> [4] J. N. Tsitsiklis. NP-Hardness of checking the unichain condition in average cost MDPs. Operations research letters, 35(3):319–323, 2007.
>
> ### Conclusion.
>
> We hope that our responses convince you of the completeness and significance of our paper and lead to an increased score.

---

> > ### Comment · Reviewer_MXst · 2023-08-18
> >
> > Thanks for addressing my question in point 4 and question 1. I believe my current rating represents the significance of the paper and I will keep my current rating.

---

### Official Review · Reviewer_mmfD · 2023-07-20

**Soundness:** 4 excellent
**Presentation:** 3 good
**Contribution:** 4 excellent
**Rating:** 7
**Confidence:** 4

**Summary:**

This paper defines the Blackwell discount factor: when the discount factor is larger than this factor, discount-optimal policies become Blackwell and average-optimal. An analytical solution for such factor is also derived. The results are also extended to the setting of sa-rectangular RMDPs.

**Strengths:**

Motivated by the original definition of Blackwell optimality, the authors propose the Blackwell discount factor, via which we can solve for the optimal policy of an average-reward MDP by solving a corresponding discounted-reward one, where the restrictive assumptions in average-reward setting is bypassed. This should be an exciting result. The writing is clear and the contents are well organized.

**Weaknesses:**

I only have some minor comments. Please see my questions below.

**Questions:**

Line 177: It should be "we use the notation...".

Line 319: It is not clear to me what "coarseness" is and how Proposition 4.3 shows this dependence.

**Limitations:**

Please see the questions above.

---

> ### Author Rebuttal · Authors · 2023-08-09
>
>
> We thank you for your time reviewing the paper.
>
> *As regards coarseness and Proposition 4.3:* Our goal is to obtain an upper bound on the Blackwell discount factor $\gamma_{\sf bw}$ for a large class of MDP instance. Proposition 4.3 shows that without any condition on the rewards $\boldsymbol{r}$ and the transition probabilities $\boldsymbol{P}$, we may have $\gamma_{\sf bw}$ as close to $1$ as we want by letting the instantaneous reward approaching $0$ (see the proof of Proposition 4.3). Inspecting the proof of Proposition 4.3, we note that $\gamma_{\sf bw} \rightarrow 1$ when the minimum difference between two instantaneous rewards (called $\epsilon$ in the proof) approaches $0$. Therefore, we study the class of MDP instance with bit-size bounded by $m \in \mathbb{N}$, for which there is a lower bound on the minimum difference between two instantaneous rewards, i.e., for which $\epsilon \geq \frac{1}{m}$. This is what we refer to as ``coarseness''. For this class of MDP instance, we can prove a uniform upper bound on $\gamma_{\sf bw}$, as shown in Theorem 4.4.
>
> We would also like to note that during this rebuttal, we have noticed that we can extend Proposition 3.4 (shortcomings of the existing definition of Blackwell optimality) and Proposition 4.3 to the case of weakly-communicating MDPs. This strengthens our contributions by showing that the shortcomings of previous work that we identify in our paper also occur under the most common structural assumptions.
>
> *Extending Proposition 3.4/Example 3.5 to unichain/ergodic MDPs:* We can extend Example 3.5 to a unichain MDP as follows: we add a transition from state $7$ to state $0$, with a reward of $0$. We also add three intermediate states from $0$ to $7$ for action $a_{1}$, so that it takes as many periods to reach state $7$ from state $0$ for the three action s$a_{1},a_{2},a_{3}$. Note that this new MDP is unichain. We refer to Figure 1 in the PDF attached to our response common to all reviewers.
>
>
> Additionally, for this new MDP instance, we have $v_{\gamma}^{a_{1}} = 1/(1-\gamma^5), v_{\gamma}^{a_{2}} = (r_{1}\gamma + r_{2}\gamma^2)/(1-\gamma^5), v_{\gamma}^{a_{3}} = (r_{4}\gamma + r_{5}\gamma^2)/(1-\gamma^5)$, which are exactly the same expression as for Example 3.5, up to the common denominator $(1-\gamma^5)^{-1}$. Therefore, we have proved that the same conclusion as Proposition 3.4 holds for unichain MDPs. We thank you for raising this interesting point; we will use the additional page available for our final version to mention this new result.
>
> *Extending Proposition 4.3 to weakly-communicating MDPs:*
> For this, we can add a deterministic transition from state $s_2$ to state $s_1$, with a reward of $0$ for action $a_{1}$ and a reward of $\epsilon$ for action $a_{2}$. We refer to Figure 2 in the PDF attached to our response common to all reviewers.
>
>  First, the MDP instance is weakly-communicating since $\{s_{1},s_{2}\}$ is strongly connected under policy $a_{2}$. In this new MDP instance, we still have $v_{\gamma}^{a_{1}} = 0$ but $v_{\gamma}^{a_{2}} = (-1+\epsilon \gamma)/(1-\gamma)$. Hence $a_2$ is Blackwell optimal when $\gamma \geq 1/\epsilon$. By choosing $\epsilon$ larger than $1$ and $\epsilon \rightarrow 1$, we obtain $\gamma_{\sf bw} \rightarrow 1$. This shows that we can extend Proposition 4.3 to unichain MDPs. We thank you for raising this interesting point; we will add this new result in the final version of our paper.
>
> ### Conclusion.
>
> We hope that our responses convince you of the completeness and significance of our paper and lead to an increased score.

---

> > ### Comment · Reviewer_mmfD · 2023-08-16
> > **Thank you!**
> >
> > Thank you for addressing my comments in detail. I will keep my score.

---

### Official Review · Reviewer_uLuR · 2023-07-21

**Soundness:** 3 good
**Presentation:** 3 good
**Contribution:** 2 fair
**Rating:** 5
**Confidence:** 3

**Summary:**

This paper studies a new class of objective for MDPs. Instead of aiming to find the policy which maximizes the long-term average reward, the authors propose a new approach to find a Blackwell optimal policy. A Blackwell optimal policy also maximizes the average reward, but Blackwell optimality has not been studied much in the literature due to the intractability. In this paper, the authors show that there exists a ‘Blackwell discount factor’ such that any policy which is discount optimal for this specific choice of discount factor is also Blackwell optimal and average reward optimal. However, finding this Blackwell discount factor is quite challenging, although the authors provide a sufficient condition for it, providing an upper bound on the Blackwell discount factor, and showing that any policy which is optimal wrt this upper bound on the discount factor is Blackwell optimal. However, this upper bound depends on properties of the MDP which are may not be known in RL. The authors also provide an extension to robust MDPs.

**Strengths:**

The notion of Blackwell optimality is interesting and indeed not studied by the RL community, so it is nice to see it being brought to attention. It is also nice that this provides a method for finding an average-optimal policy in a known MDP without assumptions on the MDP, and it is pleasing that there is an extension to robust MDPs. The paper is reasonably well written and organised.

**Weaknesses:**

It seems that Blackwell optimality has been considered in MDPs before, so the main contribution seems to be defining this Blackwell discount factor and showing solving an MDP with that discount factor leads to an average-optimal policy. From the discussion of the literature, it sounds like the existence of the Blackwell discount factor was folklore so it is nice to see it proven. However, the Blackwell discount factor itself seems difficult to find and the proposed upper bound also depends on quantities such as the maximum bit size of the MDP which I imagine is typically not known. However, there was only limited discussion of the proposed bound so I found it somewhat difficult to interpret. The bound on the discount factor also seems to scale with 1-1/(4^S)^S, so as S gets bigger this will approach 1 very quickly (indeed when I plotted it on my laptop, for S=4, it was rounded to 1). Therefore, I am concerned that for most reasonably sized MDPs, we will need to find an optimal policy with a discount factor arbitrarily close to 1 which will still be challenging and may require more assumptions, so I am not completely sure of the benefit of this approach.
Lastly, it is not clear how to translate this approach to an RL setting where we do not know the MDP. There was a bit of discussion about this, but not sufficient to understand how to actually use it in RL. There is an attempt to generalise to robust MDPs where we only know a set of transition functions, but I did not think sufficient details were provided to fully understand the usefulness of this result. I also wonder whether this work could be better suited to another venue.

**Questions:**

Can more details of calculating the bit-size of an MDP be given? In particular, it would be nice to see an example of the bit-size of an MDP (e.g. riverswim or another classical example) be given? Can this also be translated into the bound on the discount factor?
It would also be good to see a plot of the bound on the discount factor with size of the state space to see when it leads to bounds that are essentially 1.
It may also be good to have a comparison in terms of computation time/number of iterations of this method and other methods, even if there are different assumptions. More generally, it would be good to have the benefits of the proposed method clearly demonstrated.
Can the results for robust MDPs be used in the setting where we define confidence sets using data?


**Limitations:**

Limitations (e.g. what to do when we don’t know the MDP) are discussed briefly but could be discussed in more detail.

---

> ### Author Rebuttal · Authors · 2023-08-09
>
>
> Thank you for your review and suggestions. We answer your remarks and questions below.
>
> *Therefore, I am concerned that for most reasonably sized MDPs, we will need to find an optimal policy with a discount factor arbitrarily close to 1 which will still be challenging and may require more assumptions, so I am not completely sure of the benefit of this approach.*
>
> We agree that the immediate practical utility of our bounds is limited. However, we believe that this result can serve as an important foundation for future theoretical and practical implications. As you observe, our upper bound on $\gamma_{\sf bw}$ is close to one, but the number $1 -\gamma_{\sf bw}$ has a number of bits that is polynomial in the size of the MDP. Theorem 4.7 then shows that the Blackwell optimal policy can be computed in polynomial time using an LP solver. Moreover, we introduce new techniques for bounding the Blackwell optimality, which can be used in the future to further refine the upper bound on the Blackwell discount factor.
>
> *I also wonder whether this work could be better suited to another venue.*
>
> We believe this is relevant to NeurIPS, particularly because the notion of Blackwell optimality in MDPs and robust MDPs has received increased attention in the reinforcement learning community (such as Dewanto et al. (2020) and (2021), Jin et al. (2021), Tang et al. (2021), Yang et al. (2016)). Our results help to rigorously establish a fact that has been used implicitly (without proofs) in recent work. If you still think that it is not a good fit, we would appreciate suggestions for an alternative venue.
>
> *In particular, it would be nice to see an example of the bit-size of an MDP e.g. riverswim or another classical example, be given?*
>
> Thank you, that is a great suggestion. We agree that an example would make the results more approachable. For example, the bit-size for riverswim would be about 14 because the number is the largest bit-size is the reward 10,000 in the terminal state. We will add such an example in the revision.
>
> *It would also be good to see a plot of the bound on the discount factor with size of the state space to see when it leads to bounds that are essentially*
>
> We agree that plotting the discount factor as a function of an MDP parameter would be interesting. We will add such a plot in the final version of the paper.
>
> *Can the results for robust MDPs be used in the setting where we define confidence sets using data?*
>
> That is an interesting question. Yes, the results also apply to any MDP or robust MDP regardless of how this MDP is constructed. In fact, our Theorem 4.11, which bounds the Blackwell discount factor for robust MDPs is specifically for uncertainty sets based on $\ell_{1}$ or $\ell_{\infty}$ balls. These distances are used to construct uncertainty based on some data, see section 6.2.3 in [1] for healthcare applications and section 6 in [2] for methodology.
>
>
> [1] Grand-Clement, Julien, Chan, Carri W., Goyal, Vineet, et al. Robustness of Proactive Intensive Care Unit Transfer Policies. Operations Research, 2022.
>
> [2] Behzadian, Bahram, Russel, Reazul Hasan, Petrik, Marek, et al. Optimizing percentile criterion using robust MDPs. AISTATS 2021.

---

> > ### Comment · Reviewer_uLuR · 2023-08-11
> >
> > Thanks for getting back to me and clarifying some of my questions. In particular, I had missed the significance of Theorem 4.7 (perhaps this can be highlighted more in a revised version) so have raised my score in light of that. However, I am still somewhat unsure of the practical utility of the bounds.

---

### Author Rebuttal · Authors · 2023-08-09

Dear editorial teams

We would like to thank all the reviewers for their constructive feedback. We provide our answers to all reviewers individually.

We would like to note that we have extended Proposition 3.4 (shortcomings of existing definition of Blackwell optimality) and Proposition 4.3 (need for some ``coarseness" condition to bound the Blackwell discount factor $\gamma_{\sf bw}$) to the case of weakly-communicating MDPs. We present the detailed computations for each reviewer who mentioned it. We also attach here the PDF with the new weakly-communicating MDP instances for our proofs of Proposition 3.4 (Figure 1) and Proposition 4.3 (Figure 2).  This strengthens our contributions by showing that the shortcomings of previous work that we identify in our paper also occur under the most common structural assumptions.

We hope that our responses convince you of the completeness and significance of our paper and we are looking forward hearing back from the reviewers.

---

### Decision · Program_Chairs · 2023-09-21

**Decision:**

Accept (poster)

**Comment:**

This work studies an optimality objective for MDPs known as a Blackwell optimal policy. Although this concept has been studied before, in this work the authors show the existence of a Blackwell discount factor such that any policy that is discount optimal for this discount factor is also Blackwell optimal. The authors also provide sufficient conditions for this factor along with an algorithm to find it using this upper bound. The reviewers agreed this work provides a meaningful addition to the field of RL.

We would like the authors to keep in mind there was a high variability in scores among the different reviewers. Many concerns have been raised including the usefulness of the Blackwell Optimality criterion, the computational efficiency of the proposed algorithm and concerns about the utility of the bounds. It would be ideal if the authors address some of these comments in the final version of the paper.